# The mitochondrial calcium uniporter of pulmonary type 2 cells determines severity of acute lung injury

Mohammad Naimul Islam [1], Galina A. Gusarova[1], Shonit R. Das[1,4], Li Li[1], Eiji Monma[1], Murari Anjaneyulu[2], Liberty Mthunzi [1], Sadiqa K. Quadri[1], Edward Owusu-Ansah[2], Sunita Bhattacharya[3] & Jahar Bhattacharya [1,2] ✉

Acute Lung Injury (ALI) due to inhaled pathogens causes high mortality. Underlying mechanisms are inadequately understood. Here, by optical imaging of live mouse lungs we show that a key mechanism is the viability of cytosolic $Ca^{2+}$ buffering by the mitochondrial $Ca^{2+}$ uniporter (MCU) in the lung's surfactant-secreting, alveolar type 2 cells (AT2). The buffering increased mitochondrial $Ca^{2+}$ and induced surfactant secretion in wild-type mice, but not in mice with AT2-specific *MCU* knockout. In the knockout mice, ALI due to intranasal LPS instillation caused severe pulmonary edema and mortality, which were mitigated by surfactant replenishment prior to LPS instillation, indicating surfactant's protective effect against alveolar edema. In wild-type mice, intranasal LPS, or *Pseudomonas aeruginosa* decreased AT2 MCU. Loss of MCU abrogated buffering. The resulting mortality was reduced by spontaneous recovery of MCU expression, or by *MCU* replenishment. Enhancement of AT2 mitochondrial buffering, hence endogenous surfactant secretion, through *MCU* replenishment might be a therapy against ALI.

The acute immune response at environment-facing epithelial barriers provides rapid systemic protection against ambient pathogens. The lung, which is particularly prone to challenge by inhaled pathogens, develops brisk immunity to clear the pathogens and protect barrier properties of the alveolar epithelium, the site of blood oxygenation. Pneumonia due to non-resolving inflammation causes barrier injury and pulmonary edema, impairing oxygenation, and setting the stage for Acute Lung Injury (ALI), a condition associated with high mortality[1]. Mechanistic understanding is inadequate as to how innate immunity to pneumonias leads to life-threatening ALI.

The mortality risk in pneumonia is due not only to the infectious load of the inhaled organism, but also to the injurious host response that causes alveolar barrier failure. The alveolar epithelium consists of type 1 cells that form the alveolar wall, and the metabolically significant type 2 cells (hereafter, AT2). AT2 secrete surfactant that stabilizes alveolar patency and protects against pulmonary edema[2,3]. However, immunity-induced mechanisms that destabilize these alveolar protective mechanisms are not well understood.

AT2 mitochondria may be important in this regard, since mitochondrial buffering dampens elevations of the cytosolic $Ca^{2+}$ ($cCa^{2+}$), unchecked increases in which can cause barrier weakening[4]. The buffering increases mitochondrial $Ca^{2+}$ ($mCa^{2+}$), activating calcium-dependent mitochondrial dehydrogenases that augment electron flow in the mitochondrial electron transport chain, increasing mitochondrial ATP production[5]. Therefore, although evidence is lacking, mitochondrial energetics may activate surfactant secretion, protecting against pulmonary edema.

Mitochondrial buffering involves $Ca^{2+}$ flux from the cytosol to the mitochondrial matrix across the mitochondrial calcium uniporter

[1]Department of Medicine, Vagelos College of Physicians and Surgeons, Columbia University, New York, NY 10032, USA. [2]Department of Physiology and Cellular Biophysics, Vagelos College of Physicians and Surgeons, Columbia University, New York, NY 10032, USA. [3]Department of Pediatrics, Vagelos College of Physicians and Surgeons, Columbia University, New York, NY 10032, USA. [4]Deceased: Shonit R. Das. ✉e-mail: jb39@cumc.columbia.edu

(MCU)[6,7]. Studies in mice involving global[8] or tissue-specific *MCU* deletion[9], or overexpression of dominant-negative *MCU*[10] reveal a determining role of the MCU in multiple aspects of organ function, inlcuding body size determination[8], exercise tolerance[8,11,12], myocardieal infarction[9,13], pulmonary fibrosis[14], insulin secretion[15], fibroblast differentiation[16], and hepatic lipidosis[17]. Despite this extensive evidence for MCU involvement in organ function, the reported data rely on a priori genetic modifications that do not replicate natural pathogenesis. For ALI, understanding is lacking as to whether loss of the MCU, hence the resulting loss of buffering, exacerbates ALI by inhibiting surfactant secretion.

Here, we address this issue in LPS and bacterial models of lung infection, using an in situ assay of mitochondrial buffering in AT2 of live alveoli by means of real-time confocal microscopy[18]. We could thereby, determine the dynamic $Ca^{2+}$ responses in the cytosol and mitochondria in relation to the progression of acute immunity. Our findings indicate that by the first post-infection day, there was marked depletion of both message and protein levels of AT2 MCU, together with loss of buffering and surfactant secretion. Unexpectedly, sustained MCU depletion promoted mortality, which could be mitigated by MCU repletion, revealing the critical role of AT2 MCU in immune alveolar injury.

## Results

### The MCU and alveolar homeostasis

We induced ALI by exposing mice to intranasally instilled LPS or *Pseudomonas aeruginosa* (PA), as we indicate below. By live confocal microscopy, we assayed MCU function in AT2 to determine the effect of mitochondrial $Ca^{2+}$ buffering on surfactant secretion. We identified AT2 as cells that stained for surfactant-containing lamellar bodies (LBs) and for surfactant protein B (Supplementary Fig. 1a)[19]. By alveolar micropuncture, we loaded the alveolar epithelium with the dyes, rhod2 and fluo4 to quantify mitochondrial ($mCa^{2+}$) and cytosolic ($cCa^{2+}$) calcium, respectively, as we described[20,21]. To confirm compartmental distribution of the dyes, we gave alveolar microinfusion of a mild detergent, which released fluo4, but not rhod2 from the epithelium, affirming that fluo4 was localized to the cytosol and rhod2 to the mitochondrial matrix[22] (Supplementary Fig. 1b, c). Alveolar stretch caused by a 15-second lung hyperinflation induced transient $cCa^{2+}$ oscillations in AT2, but no increase of mean $cCa^{2+}$ (Fig. 1a, b). Concomitantly, both the mean and the oscillation amplitude of mitochondrial $Ca^{2+}$ increased (Fig. 1a, b and Supplementary Movie 1), affirming onset of mitochondrial buffering that protected against $cCa^{2+}$ increases.

We tested these responses in *MCU^{F/F}:SPC-rtTa-tetO-Cre* mice (*rtTa^{MCU−/−}*) in which we induced pre-partum Cre recombination in the alveolar epithelium (AE)[21,23] by doxycycline treatment. *rtTa^{MCU−/−}* mice lacked MCU in AT2 (Supplementary Fig. 1d–f). For control, we withheld doxycycline in *rtTa-tetO-Cre* expressing mice. We imaged alveoli of equal diameters in control and *rtTa^{MCU−/−}* mice. The hyperinflation-induced mitochondrial responses were inhibited pharmacologically by alveolar treatment with the MCU blocker, ruthenium red and they were absent in the *rtTa^{MCU−/−}* mice (Supplementary Fig. 1g and Fig. 1b), confirming the MCU role in the buffering. Further, loss of buffering caused the expected increase of mean $cCa^{2+}$ (Fig. 1b). The inhibitor of store $Ca^{2+}$ release, xestospongin C blocked all $Ca^{2+}$ increases (Supplementary Fig. 1g), consistent with the notion that MCU-dependent $mCa^{2+}$ increases were due to mitochondrial entry of store-released $Ca^{2+}$. Surfactant secretion in intact alveoli is assayed by labeling LBs with the fluorescent dye, Lysotracker Red (LTR), then detecting the time-dependent loss of the fluorescence to a secretion stimulus such as hyperinflation[18,19] (Supplementary Fig. 1h–j and Fig. 1c). However, this response was absent in the *rtTa^{MCU−/−}* mice (Fig. 1c), as also in mice treated with ruthenium red, indicating that blocking mitochondrial $Ca^{2+}$ entry blocked surfactant secretion (Supplementary Fig. 1i, j).

These findings indicated that the AT2 MCU is a major determinant of surfactant secretion.

To rule out possible toxicity of the rtTa-tetO system[24], we bred *MCU^{F/F}:SPC-Cre-ERT2* mice (*ERT2^{MCU−/−}*) as an alternative system for activation of Cre recombinase in AT2. We affirmed that the hyperinflation-induced AT2 responses, namely mitochondrial calcium buffering and surfactant secretion, were absent (Fig. 1d, e). Thus, our key findings could be replicated by two different strategies of Cre recombinase activation, ruling out non-specific effects of Cre recombination as a complicating factor.

### ALI causes MCU loss

We determined time dependent responses of AT2 mitochondria in the LPS-induced mouse model of ALI[18,21]. Since mouse strains differ in susceptibility to LPS[18,21], we strain adjusted the LPS dose as we indicate (Supplementary Table 3). A nonlethal LPS dose (1 mg/kg) in the *Swiss Webster* strain markedly increased leukocyte counts in the bronchioalveolar lavage (BAL) within Day 1 (Supplementary Fig. 2a). The counts recovered to baseline by Day 5 (Supplementary Fig. 2a). On Day 1, surfactant secretion was repressed (Fig. 2a), and extravascular lung water, a measure of pulmonary edema, increased (Supplementary Fig. 2b).

In lungs derived from these LPS-treated mice, the AT2 physiological responses, namely $mCa^{2+}$ increase and surfactant secretion, were present at 4 h post-LPS, but repressed at 24 h (Fig. 2a, b). Immunoblots of AT2 mitochondria indicated that MCU expression progressively decreased and was significantly lower than baseline by 16 h (Fig. 2c). MCU RNA decreased to about 50% of baseline levels (Fig. 2d). These findings indicated that LPS deceerased AT2 MCU and mitochondrial buffering by Day 1.

We applied an optogenetic approach as an alternative approach for increasing $cCa^{2+}$ [25]. We expressed the light sensitive, cation channel, channelrhodhopsin-2 (ChR2) in the alveolar epithelium (Supplementary Fig. 2c). Excitation of ChR2 by blue light increased AT2 mitochondrial $Ca^{2+}$ markedly less in LPS- than PBS-treated lungs (Supplementary Fig. 2d, e and Fig. 2e). Hence, the hyperinflation and optogenetics approaches together denoted loss of MCU-induced buffering in AT2 following LPS treatment.

At Day 1 post-LPS, the mitochondrial outer membrane proteins, TOM-20 and VDAC, and the matrix protein HSP60 were well expressed (Supplementary Fig. 2f, g). The proteins of the mitochondrial electron transport chain (ETC) were also well expressed and their activities were not diminished (except for a small decrease in Complex II activity) (Supplementary Fig. 2h, i). Therefore, the MCU loss was not due to a non-specific loss of mitochondrial proteins.

The mitochondrial membrane potential informs mitochondrial fitness. Mitochondrial depolarization stabilizes PINK1 on the outer membrane. PINK1 recruits the E3-ubiquitin ligase, parkin, which initiates mitophagy[26]. To determine whether the MCU impacted mitochondrial potential, we loaded the alveolar epithelium with the potentiometric dye, TMRE, fluorescence of which decreases with mitochondrial depolarization[4,22]. Our findings indicate that despite the loss of the MCU, TMRE fluorescence was unchanged (Supplementary Fig. 2j, k), indicating that the AT2 mitochondria were not depolarized, hence they were unlikely to be mitophagy targets of the PINK1-parkin mechanism. To test this hypothesis, we carried out immunoblots on mitochondria derived from AT2 24 h after LPS treatment. These studies failed to reveal association of parkin (Supplementary Fig. 2l), suggesting that the MCU loss did not induce a mitophagy signal.

To evaluate the effect of inflammation severity, we exposed the *Swiss Webster* mice to intranasal LPS at nonlethal (1 mg/kg) or lethal (10 mg/kg) doses (Supplementary Table 3)[18]. For the nonlethal dose, the MCU loss on Day1 was followed by recovery of MCU expression (Fig. 2f). By contrast, for the lethal dose the expression did not recover (Fig. 2f). Similarly, although both doses caused loss of body weight by

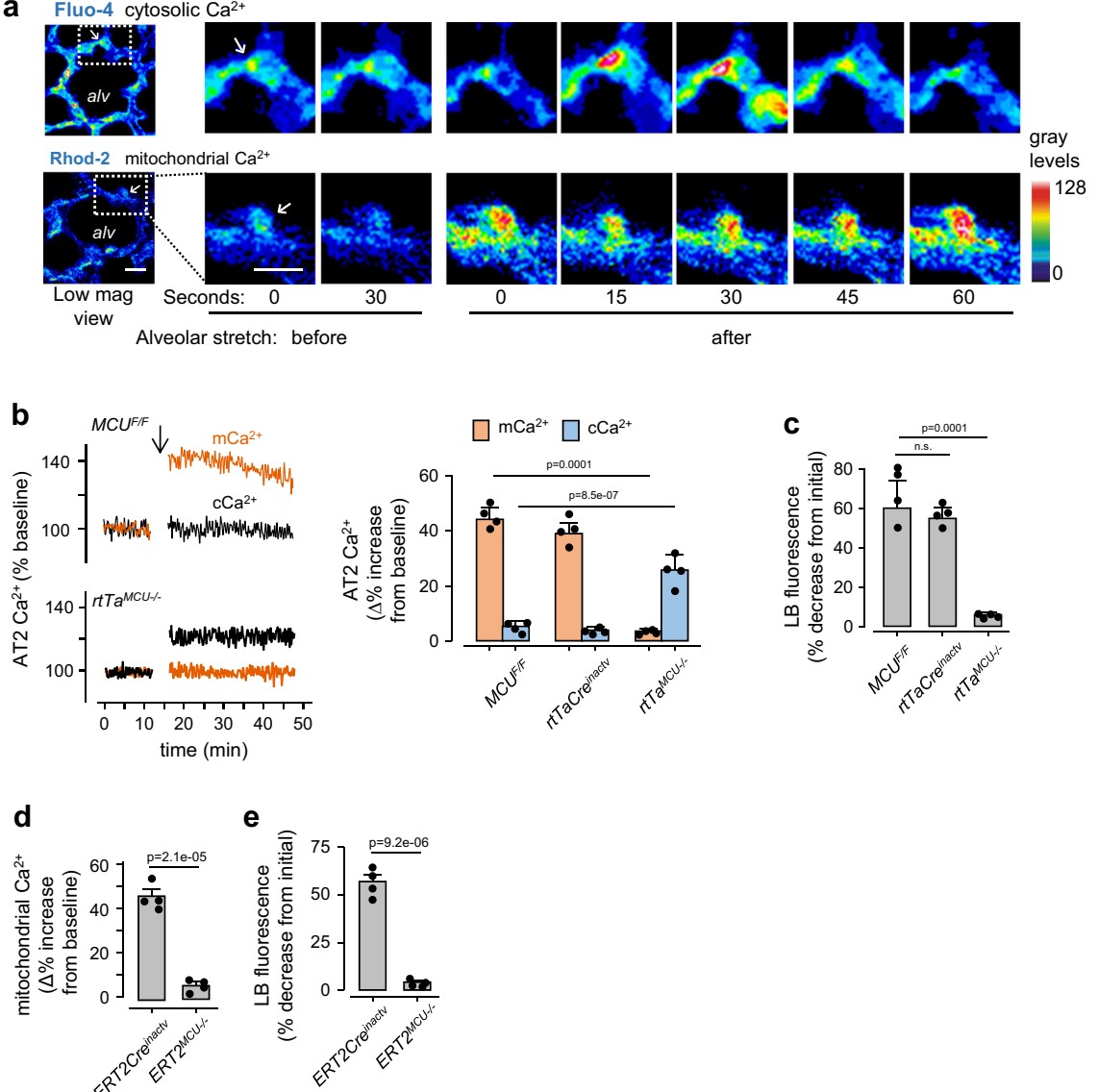

**Fig. 1 | In situ AT2 mitochondrial responses to hyperinflation. a** Confocal images of a live alveolus (*alv*) show AT2 (*arrows*) in a septum of the alveolar wall, as identified by Lysotracker Red (LTR) staining. Images show time dependent $Ca^{2+}$ responses in the cytosol ($cCa^{2+}$) and mitochondria ($mCa^{2+}$) of selected AT2 to a single 15-second hyperinflation induced by increasing airway pressure from 5 to 15 $cmH_2O$. $Ca^{2+}$ dyes were given by alveolar microinfusion. $Ca^{2+}$ increases are denoted by increases of pseudocolored gray levels. Images were obtained at airway pressure of 5 $cmH_2O$. Scale bar, 10 μm. Repeat images were taken at 3 different locations per lung in 4 lungs per group. **b** Tracings from an experiment and group data show AT2 $Ca^{2+}$ responses to hyperinflation (*arrow*). *MCU^F/F^*, mice floxed for the MCU; *rtTaCre^inactv^*, mice expressing rtTA-Cre not exposed to doxycycline; *rtTa^MCU−/−^*, mice lacking the MCU in the alveolar epithelium. Bars: mean ± SEM. *n* = 4 mice per

group. Groups were compared using one-way ANOVA with Bonferroni correction. **c** Group data show hyperinflation-induced surfactant secretion as quantified by loss of AT2 fluorescence of the lamellar body dye, lysotracker red (LTR). Negligible fluorescence loss indicates negligible surfactant secretion. Bars: mean ± SEM. *n* = 4 lungs for each bar. Groups were compared using one-way ANOVA with Bonferroni correction. *n.s.*, not significant. **d**, **e** Group data are determinations of alveolar hyperinflation-induced mitochondrial $Ca^{2+}$ (*d*) and surfactant secretion (*e*) responses in the indicated groups. We crossed the MCU^f/F^ mice with mice bearing an inducible Cre recombinase under the control of an SPC promoter (SPC-Cre-ERT2). Post-partum Cre activation by tamoxifen (i.p.) caused AT2 MCU deletion in these mice. ERT2^MCU−/−^, tamoxifen treated; ERT2Cre^inactv^, tamoxifen untreated. Bars: mean ± SEM. *n* = 4 lungs for each bar, p-values are for two-tailed Student's *t*-test.

Day 1, the weight recovered for the nonlethal, but not the lethal dose (Fig. 2g). These findings suggested that adequacy of AT2 MCU expression was a determinant of survival.

Since the effects of purified LPS might not be entirely representative of those induced by Gram-negative bacteria, we infected lungs by intranasal instillation of *Pseudomonas aeruginosa* (PA) at a high and a low dose to induce dose-dependent increases of leukocyte counts in the bronchioalveolar lavage by Day 1 (Fig. 2h). Concomitantly, the loss of AT2 MCU expression was also dose-dependent (Fig. 2i). These findings indicated that the lung's immune response caused a dose dependent loss of AT2 MCU expression.

## Rescue strategies against LPS-induced mortality

In *ERT2^MCU−/−^* mice *MCU* was specifically deleted in AT2 through post-partum tamoxifen activation of the Cre recombinase (Fig. 3a). To determine the effects of *MCU* addback, we transduced *MCU* in the knockout mice by intranasal plasmid delivery in *ERT2^MCU−/−^* mice. Thus, MCU expression, which was absent in AT2 of the knockout mice, was reinstated by the transduction (Fig. 3a). An LPS dose that caused moderate mortality in Cre-inactivated, littermate controls, induced severe mortality in *ERT2^MCU−/−^* mice (Fig. 3b). Similarly, the high mortality following LPS was also evident in *rtTa^MCU−/−^* mice (Supplementary Fig. 3a). Thus, loss of MCU in AT2 mitochondria enhanced mortality in a strain-

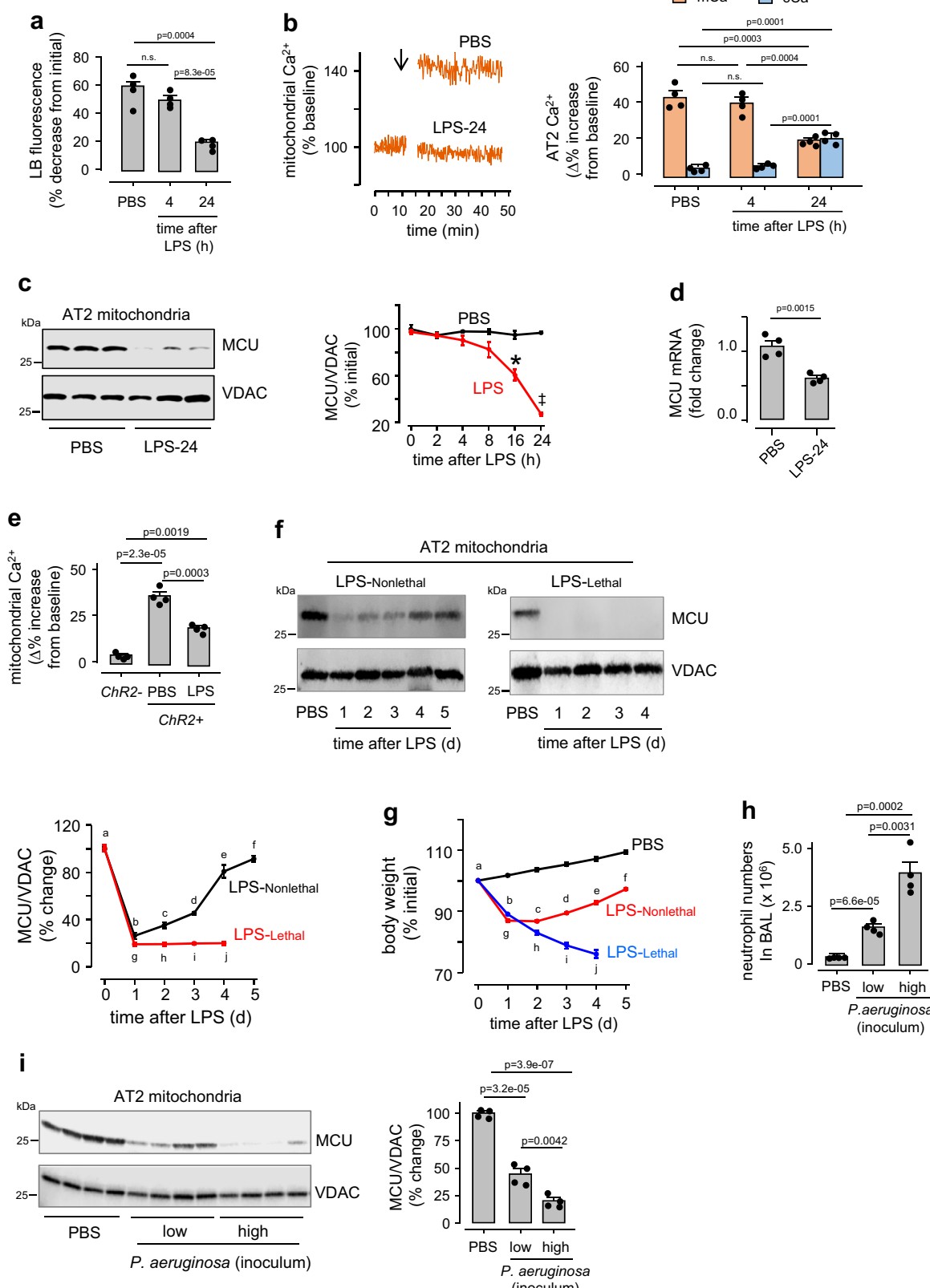

and Cre-independent manner. Further, *MCU* transduction markedly abrogated the mortality (Fig. 3b). Taken together, we interpret that loss of *MCU* in AT2 was of systemic homeostatic significance.

In our previous studies we noted that wild-type mice of the *Swiss Webster* strain[18] are more susceptible to LPS-induced mortality as compared with other strains[21]. To determine whether the susceptibility

is reversed by *MCU* enrichment, we transfected *MCU* in *Swiss Webster* mice to increase AT2 MCU expression (Supplementary Fig. 3b, c). A subsequent intranasal instillation of LPS decreased MCU expression in 24 h (Fig. 3c), as before. However, despite this decrease, the expression remained sufficiently high (60% of baseline) that buffering was retained (Fig. 3c, d). Although LPS caused the expected decrease of

**Fig. 2 | LPS effects on AT2 mitochondria. a** Group data show quantifications of hyperinflation-induced surfactant secretion for the indicated groups. LPS was given at nonlethal dose (1 mg/kg, Supplementary Table 3). Bars: mean ± SEM. $n = 4$ lungs each bar. Groups were compared using one-way ANOVA with Bonferroni correction. *n.s.*, not significant. **b** Tracings from a single experiment and group data show hyperinflation (*arrow*) -induced AT2 mitochondrial responses following the indicated intranasal instillations. LPS was given at nonlethal dose (1 mg/kg, Supplementary Table 3). *LPS-24*, 24 h after intranasal LPS. Repeated in 4 independent experiments per group. **c** Immunoblots are shown for mitochondrial calcium uniporter (MCU) in AT2 mitochondria derived 24 h after indicated treatments. LPS was instilled at a nonlethal dose in *Swiss Webster* mice. *VDAC*, voltage dependent anion channel. Tracings show densitometry quantification of MCU/VDAC ratio. Replicated 4 times. *$p = 0.004$ and ‡$p = 3.4e{-}06$ versus corresponding PBS. p-values are for two-tailed Student's t-test. **d** Data are quantifications of the MCU RNA in freshly isolated AT2 from mice given the indicated intranasal instillations. Lungs were excised and mRNA extracted at indicated time points after instillations of nonlethal LPS. RNA values were normalized against actin. Bars: mean ± SEM. $n = 4$ lungs for each bar, p-value is calculated by two-tailed Student's t-test. **e** Group data show Ca$^{2+}$ responses of AT2 mitochondria in live alveoli to a 10-second channelrhodopsin (*ChR2*) activation. *ChR2-GFP* plasmid was given by intranasal instillation in liposomal complexes. Lungs were excised 48 h after instillation. *ChR2+* and *ChR2-* are respectively, data for AT2 that were positive or negative for GFP fluorescence in the same lung. Bars: mean ± SEM. n = 4 lungs per group. Groups were compared using one-way ANOVA with Bonferroni correction. **f** MCU immunoblots in AT2 mitochondria and densitometric quantification of MCU/VDAC ratio after intranasal instillations of nonlethal (*LPS-Nonlethal*, 1 mg/kg, *left*) and lethal (*LPS-Lethal*, 10 mg/kg, *right*) LPS in *Swiss Webster* mice. PBS was instilled on Day 0. Supplementary Table 3 gives details of LPS doses. Data are mean ± SEM. $n = 3$ mice for each time point. p-values for within group comparisons for Day 0 (*a*) versus the corresponding time points (*b-j*) are respectively, 1.1e{-}05, 2.6e{-}05, 5.6e{-}06, 0.02, 0.04, 1.08e{-}05, 1e{-}05, 8.3e{-}06 and 1.2e{-}05. The p-values for between group comparisons, namely *h versus c, i versus d, j versus e*, are respectively, 0.005, 3.3e{-}05 and 0.003. Groups were compared using one-way ANOVA with Bonferroni correction. **g** Group data show changes in body weight at indicated time points following intranasal PBS (*black*), nonlethal (LPS-Nonlethal, *red*) and lethal (LPS-Lethal, *blue*) LPS instillations in *Swiss Webster* mice. Data are mean ± SEM. $n = 4$ mice for each time point. *p*-values for within group comparisons for Day 0 (*a*) versus the corresponding time points (*b-j*) are respectively, 9.1e{-}15, 3.2e{-}17, 8.7e{-}07, 5.6e{-}11, 7.8e{-}06, 1.2e{-}22, 3.8e{-}19, 1.6e{-}16, and 8.6e{-}16. The p-values for between group comparisons, namely *h versus c, i versus d, j versus e*, are respectively, 7.7e{-}05, 6e{-}06, and 1.1e{-}08. Groups were compared using one-way ANOVA with Bonferroni correction. **h** Group data are determinations of neutrophil counts in bronchoalveolar lavage 24 h after intranasal *P. aeruginosa* instillations. The low and high inoculum concentrations were $1 \times 10^5$ and $1 \times 10^6$ colony-forming units (CFU), respectively. Bars: mean ± SEM. $n = 4$ mice per group. Groups were compared using one-way ANOVA with Bonferroni correction. **i** Immunoblots are shown for the MCU in AT2 mitochondria derived 24 h after indicated instillations. Group data show densitometry quantification of MCU/VDAC ratio. Bars: mean ± SEM. $n = 4$ mice per group. Groups were compared using one-way ANOVA with Bonferroni correction.

ATP in controls[18], and the expected high mortality[18], *MCU*-transfection protected ATP production (Fig. 3e), surfactant secretion (Fig. 3f), and survival (Fig. 3g). Together, these studies indicated that MCU overexpression ensured adequate protection of AT2 buffering and surfactant secretion, thereby protecting survival.

By contrast, post-LPS *MCU* transfection, which we initiated 12 h after LPS instillation, did not increase MCU expression in AT2 mitochondria, although the expression was evident in the AT2 cytosol (Fig. 3h). Since the increased cytosolic expression attests to success of transfection, we are uncertain as to why the mitochondrial expression failed. Failure of mitochondrial import of the MCU remains a possibility. Together, these studies indicated that the transfection approach for enriching AT2 MCU was likely to succeed as a protective (pre-ALI), but not a therapeutic (post-ALI) strategy for ALI.

Clements proposed that deficiency of secreted surfactant increases the alveolar air-liquid surface tension, causing as a result, decrease of the interstitial pressure, hence increase of microvascular filtration and edema[27]. We tested the Clements hypothesis in *ERT2$^{MCU-/-}$* mice that failed to secrete surfactant in response to hyperinflation. In these knockout mice, LPS caused greater increases of lung water (Fig. 3i), and severer mortality than controls (Fig. 3b), although the increases in BAL protein levels were similar (Fig. 3j). The LPS-induced lung injury was markedly abrogated by a single intranasal dose of purified surfactant (Curosurf) given 1 h prior to LPS instillation (Fig. 3b, i). Thus consistent with Clements' proposal, LPS-induced mortality in *ERT2$^{MCU-/-}$* mice resulted from severe edema and respiratory failure due to absent surfactant secretion.

As a possible therapeutic approach for enriching AT2 MCU in ALI, we considered a mitochondrial transfer strategy. Mitochondrial transfer from bone marrow-derived mesenchymal stromal cells (BMSCs) to the alveolar epithelium protects against ALI[18]. However, the protective mechanisms are not understood. To determine the role of the AT2 MCU in this protection, we transfected BMSCs with plasmids encoding GFP-tagged, wild-type *MCU* (*pMCUwt*) (Supplementary Fig. 4a), or a mutant *MCU* (*pMCUmt*) that inhibits mitochondrial Ca$^{2+}$ entry[28]. We intranasally instilled BMSCs expressing the *MCU* plasmids 4 h after instilling LPS. In lungs obtained from these mice 24 h after LPS, we confirmed that BMSC mitochondria were transferred to AT2 (Supplementary Fig. 4b, c), and that the transfer increased MCU expression in AT2 (Fig. 4a). Overexpression of *MCUwt* protected MCU function and abrogated inflammation, increasing survival (Fig. 4b-e). Overexpression of *MCUmt* failed to achieve the protections (Fig. 4b-e). Taking our findings together, we conclude that primary lack of the AT2 MCU was sufficient to tilt the inflammatory outcome towards ALI, and that rescue of MCU function by mitochondrial transfer protected inflammation resolution and survival.

## LPS-induced mitochondrial H$_2$O$_2$ degrades MCU of AT2 mitochondria

To determine the role of mitochondrial H$_2$O$_2$ production in these responses[29,30], we expressed the matrix-targeted, H$_2$O$_2$ sensor roGFP[31]. Transfection of roGFP by intranasal delivery of liposomal plasmid avoids potential difficulties due to mitochondrial uptake of diffusible H$_2$O$_2$ sensing dyes[32]. Intranasal LPS instillation progressively increased H$_2$O$_2$ in AT2 mitochondria (Supplementary Fig. 5a and Fig. 5a). To block this response, we expressed catalase in AT2 mitochondria by crossing inducible *SPC-rtTa-Cre* mice with mice expressing mitochondria-targeted human *catalase* downstream of a floxed STOP codon[33]. Post-partum induction of *SPC-Cre* caused catalase expression in AT2 mitochondria of these mice (*AT2$^{CAT+/+}$*), as confirmed by alveolar immunofluorescence and immunoblot (Supplementary Fig. 5b, c). LPS treatment of *AT2$^{CAT+/+}$* mice failed to increase mitochondrial H$_2$O$_2$, or induce loss of AT2 MCU (Fig. 5b, c). Accordingly, MCU function was retained (Fig. 5d, e), alveolar inflammation was mitigated (Fig. 5f), and mortality was reduced (Fig. 5g). Mice pre-treated with the mitochondria-specific anti-oxidant, MitoQ were also protected from the MCU loss (Supplementary Fig. 5d), indicating efficacy of pharmacalogic intervention. Together, these findings indicated that LPS-induced increase in AT2 mitochondrial H$_2$O$_2$ was a critical mediator of the MCU loss.

We considered the possibility that mitochondrial H$_2$O$_2$ might induce mitochondrial fragmentation, and thereby impair mitochondrial Ca$^{2+}$ uptake[34]. Mitochondrial H$_2$O$_2$ can activate mitochondrial fragmentation by inducing dynamin-related protein-1 (Drp1)[35,36]. Since mitochondrial distribution has not been determined in intact alveolar epithelium, we imaged AT2 in alveoli of *PhAM floxed:E2a-Cre* mice that globally express the mitochondria-targeted fluorescent protein, Dendra-2[37] (Supplementary Fig. 6a, and Fig. 6a). AT2 mitochondria aggregated at peri-nuclear poles, co-existing with surfactant-

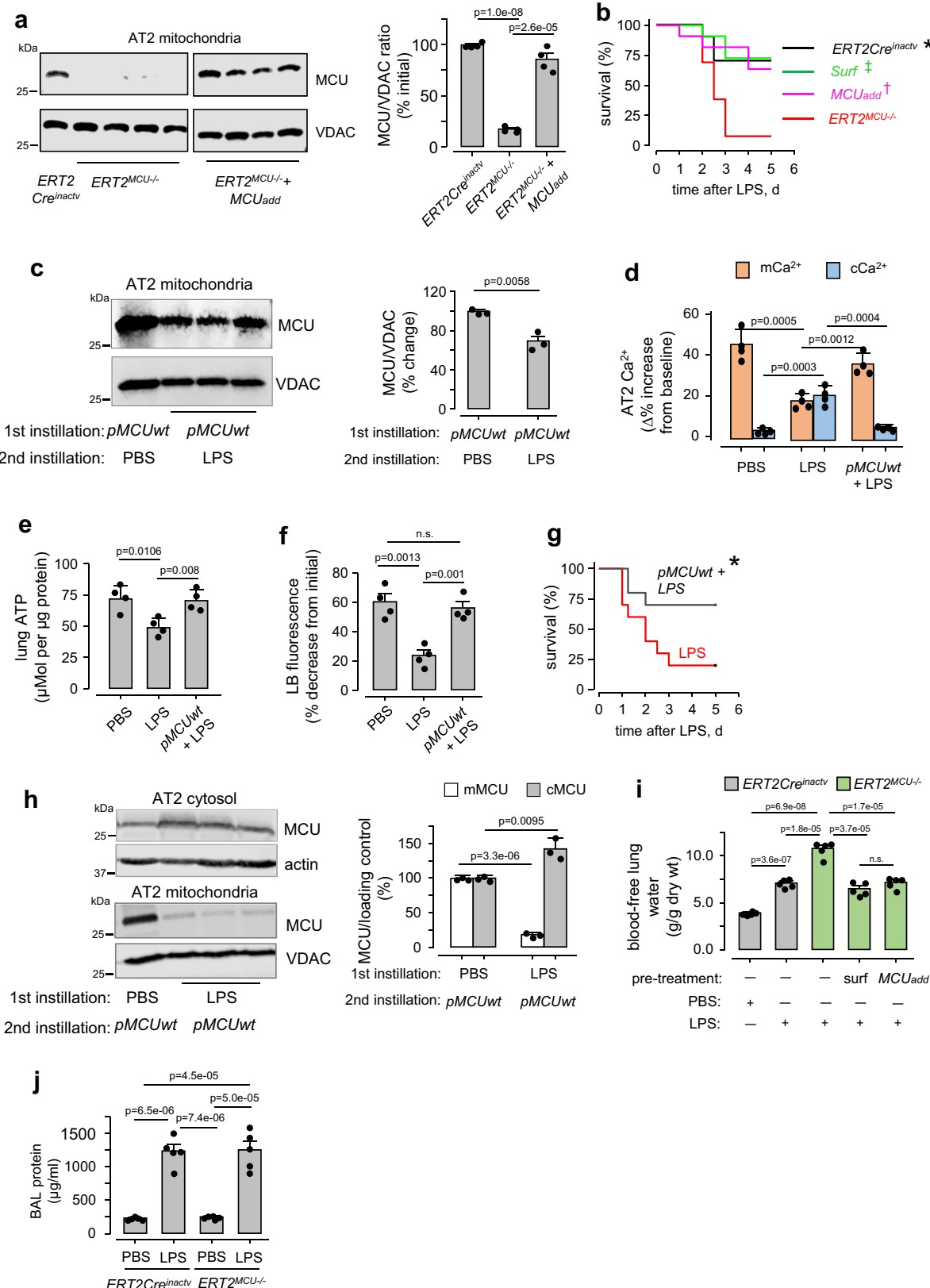

containing lamellar bodies. This polarized aggregation was lost by Day 1 after LPS, as fragmented mitochondria distributed throughout the cytosol (Fig. 6a, b). However, following intranasal delivery of the Drp1 inhibitor, Mdivi-1[38], LPS failed to disrupt the polarized mitochondrial distribution (Fig. 6a, b). Thus, Drp1 inhibition blocked LPS-induced mitochondrial disaggregation.

We confirmed that Drp1 was activated, as revealed by Drp1 phosphorylation at Ser-616 through immunoblots of AT2 mitochondria with a specific antibody[39] (Fig. 6c). Immunoblots also revealed Drp1 localization to mitochondria in floxed littermate controls, but not in $AT2^{CAT+/+}$ mice (Fig. 6d). Thus Drp1 activation was $H_2O_2$ dependent. To evaluate this mechanism further, we induced AT2-specific *Drp1*

**Fig. 3 | Effects of wild-type *MCU* overexpression in alveolar epithelium.**
**a** Immunoblots are for MCU in AT2 mitochondrial fractions derived from indicated strains. We crossed the *MCU*[F/F] mice with mice bearing an inducible Cre recombinase under the control of an SPC promoter (*SPC-Cre-ERT2*). Post-partum Cre activation by tamoxifen (*i.p.*) caused AT2 MCU deletion in these mice. *MCU*[F/F], *MCU* floxed; *ERT2*[MCU−/−], tamoxifen-treated; *ERT2Cre*[inactv], tamoxifen untreated. To add-back MCU in *ERT2*[MCU−/−] mice (*ERT2*[MCU−/−] + *MCUadd*), we i.n. instilled plasmid encoding for the wild-type MCU (*pMCUwt*) in liposomal complex. Lungs were excised 48 h after plasmid instillation. Group data show densitometry quantification of MCU/VDAC ratio. Bars: mean ± SEM. *n* = 4 mice per group. Groups were compared using one-way ANOVA with Bonferroni correction. **b** Survival plots in mice exposed to a moderate LPS dose (30 mg/kg). *Surf*, surfactant (Curosurf, 2 ml/kg body weight) intranasally instilled 1 h prior to LPS in *ERT2*[MCU−/−] mice; *MCUadd*, MCU addback 24 h prior to LPS in *ERT2*[MCU−/−] mice. *N* = 13 mice in the *ERT2*[MCU−/−] group and 11 in the other groups. \*$p = 0.017$, ‡$p = 0.006$, and †$p = 0.014$ versus *ERT2*[MCU−/−]. *p*-values are calculated by Log-Rank test. **c** Immunoblots and densitometric quantifications of AT2 mitochondria from lungs given the indicated intranasal instillations. Liposome-complexed *pMCUwt* plasmid was given by intranasal instillation. Instillation sequences were: PBS followed 8 h later by *pMCUwt* for the control group, and *pMCUwt* followed 24 h later by instillation of nonlethal LPS (1 mg/kg) for the treated group. Lungs were removed for analyses 48 h after the first instillations. Bars: mean ± SEM. *n* = 3 lungs for each bar, *p*-value is for two-tailed Student's *t*-test. **d**–**f** Responses shown are for Ca²⁺ (*d*), lung ATP (*e*) and surfactant secretion (*f*) obtained 24 h after intranasal instillations of PBS or nonlethal LPS (1 mg/kg) in wild-type or in *pMCUwt* expressing mice. *pMCUwt*-complexed liposomes were intranasally instilled 24 h before instillations of LPS. In *d*, group data are AT2 Ca²⁺ responses to hyperinflation. In *e*, lung ATP was determined by a colorimetric method. In *f*, surfactant secretion was quantified in terms of loss of AT2 fluorescence of lysotracker red (LTR). Bars: mean ± SEM. *n* = 4 lungs for each bar. Groups were compared using one-way ANOVA with Bonferroni correction. *n.s.*, not significant. **g** Survival plots are for *Swiss Webster* mice following instillations of LPS (10 mg/kg) followed 24 h later with PBS (*LPS*) or *pMCUwt* followed 24 h later with LPS (*pMCUwt*). *n* = 10 mice in each group, \*$p = 0.031$ versus LPS. *p*-value is calculated by Log-Rank test. **h** Immunoblots and densitometric quantifications of AT2 cytosol (upper) and mitochondria (lower) from lungs given the indicated intranasal instillations. Liposome-complexed *pMCUwt* plasmid was given by intranasal instillation 12 h after instillations of PBS or nonlethal LPS (1 mg/kg). Lungs were removed for analyses 48 h after the first instillations. Bars: mean ± SEM. *n* = 3 lungs for each bar, *p*-values are for two-tailed Student's *t*-test. **i, j** Group data are determination of extravascular lung water (*i*) and BAL protein (*j*) after i.n. instillation of PBS or LPS (30 mg/kg) in *ERT2cre*[inactv] and the *ERT2*[MCU−/−] mice. In *i*, pre-treatments were: *ERT2*[MCU−/−] mice given i.n. surfactant (2 ml/kg body weight) or *pMCUwt* plasmid, 1 and 24 h before LPS, respectively. Quantifications were made 72 h after PBS or LPS instillations. In *j*, protein determination in BAL were made 72 h after PBS or LPS instillations. Bars: mean ± SEM. *n* = 5 lungs for each bar. Groups were compared using one-way ANOVA with Bonferroni correction. *n.s.*, not significant.

deletion through post-partum Cre recombination in *Drp1*[F/F]:*SPC-rtTa-Cre* mice (*AT2*[Drp1−/−]). We confirmed the deletion by immunoblot of isolated AT2 (Supplementary Fig. 6b). In *AT2*[Drp1−/−] mice, there was no LPS-induced MCU loss (Fig. 7a). The MCU loss was also inhibited in mice with AT2 expression of a Drp1 mutant that lacks GTPase activity[40,41] (Fig. 7b), or after pre-treatment with Mdivi-1 (Supplementary Fig. 6c). These findings indicated that Drp1activation caused the MCU loss.

Since MCU loss inhibited surfactant secretion, we evaluated the secretion in LPS-treated *AT2*[Drp1−/−] mice. Since MCU was not lost in these mice, surfactant secretion was robust (Fig. 7c). Since $H_2O_2$ production was upsteam of the Drp1 activation, the LPS-induced $H_2O_2$ increase was present in *AT2*[Drp1−/−] mice (Fig. 7d). Thus, we show that LPS exposure induced a chain of events in AT2 mitochondria, in which $H_2O_2$-induced Drp1 activation caused mitochondrial fragmentation and loss of MCU.

### Factors contributing to mitochondrial responses in AT2

In LPS-treated lungs episodic AT2 cCa²⁺ increases occur because of Ca²⁺ conduction through connexin 43 (Cx43)-containing gap junctions (GJs) in the alveolar epithelium (AE)[42,43]. *Cx43*[F/F]:*SPC-rtTa-Cre* mice lack Cx43 in the AT2 (*AT2*[Cx43−/−]) (Supplementary Fig. 7a). As opposed to floxed littermate controls, in *AT2*[Cx43−/−] mice LPS failed to decrease MCU (Fig. 8a) and mitochondrial buffering was robust (Fig. 8b). These findings indicated that the presence of AE GJs supported the MCU loss.

It is proposed that LPS-induced activation of Toll-like receptor (TLR4) translocates the TLR-signaling adaptor molecule, TRAF6 to mitochondria[44]. TRAF6 forms a complex with Complex I assembly factor, ECSIT (evolutionarily conserved signaling intermediate in Toll pathways), augmenting mitochondrial $H_2O_2$ formation[45]. We tested this possibility in mice with heterozygous knockout of ECSIT (*ECSIT*[+/−]) in which ECSIT expression is 50% of wild type[45]. In *ECSIT*[+/−] mice, the LPS-induced MCU loss was markedly mitigated (Fig. 8c). Taken together, our findings indicate that Cx43-containing AT2 GJs and mitochondrial ECSIT contributed to the MCU loss.

### Discussion

We report here that viability of the AT2 MCU determines ALI severity. Thus, deletion of AT2 *MCU* markedly exacerbated LPS-induced mortality. Exposing mice to intranasal LPS, or *P. aeruginosa* progressively decreased MCU expression. Survival depended on spontaneous or experimental MCU replenishment, which restored mitochondrial buffering in AT2, mitigating inflammation and abrogating mortality.

Thus, the MCU was revealed as critical for sustaining AT2 mitochondrial buffering capacity that protected survival during immune stress.

Our findings mechanistically link AT2 mitochondria and lung hyperinflation, the known physiological stimulus for surfactant secretion[42,46]. We reported that hyperinflation-induced cCa²⁺ increases are conveyed by gap junctional communication to AT2[42]. We now show that the subsequent MCU-dependent buffering caused at least two protective effects, namely first, protection against potentially deleterious effects of elevated Ca²⁺; second, increase of mCa²⁺ that boosted mitochondrial ATP production to activate surfactant secretion.

To determine adverse effects of impaired buffering, and possibly of Cre recombination, we knocked out *MCU* in two groups of mice, namely in *rtTa*[MCU−/−] and *ERT2*[MCU−/−] mice, by means of two different drivers of Cre recombinase. Buffering was similarly impaired in each group, ruling out non-specific effects of recombination[24]. Buffering was also impaired in LPS-treated mice in which there was loss of MCU. In each of these groups the protective effects of buffering were lost. Hence, AT2 mitochondrial buffering was the critical mechanism that protected alveoli against the potential pathogenicity of elevated cCa²⁺ and surfactant secretion failure.

Based on the Clements hypothesis, namely that surfactant deficiency causes pulmonary edema[27], exogenous surfactant therapy has been used to treat clinical syndromes of ALI. However for reasons not well understood, exogenous surfactant has been beneficial in the neonatal, but not the adult respiratory distress syndrome[47]. Since a rigorous experimental evaluation of the Clements hypothesis is lacking, we tested the hypothesis in *ERT2*[MCU−/−] mice, which showed evidence of failed surfactant secretion, providing an opportunity to evaluate the effects of surfactant replenishment. However, since LPS treatment resulted in severe pulmonary edema in these knockout mice, we instilled surfactant prior to giving LPS to avoid the surfactant neutralizing effect of edema. Notably, the pre-LPS surfactant replenishment caused marked protection against the LPS-induced lethality, providing definitive evidence that pre-existing failure of surfactant secretion contributed to the lethal edema. These findings support the Clements hypothesis. They show further that MCU-dependent buffering was the critical mechanism that protected surfactant secretion and survival following LPS challenge.

LPS caused the expected hyperpermeability of the alveolar barrier, as indicated by increases in the BAL leukocyte count and protein

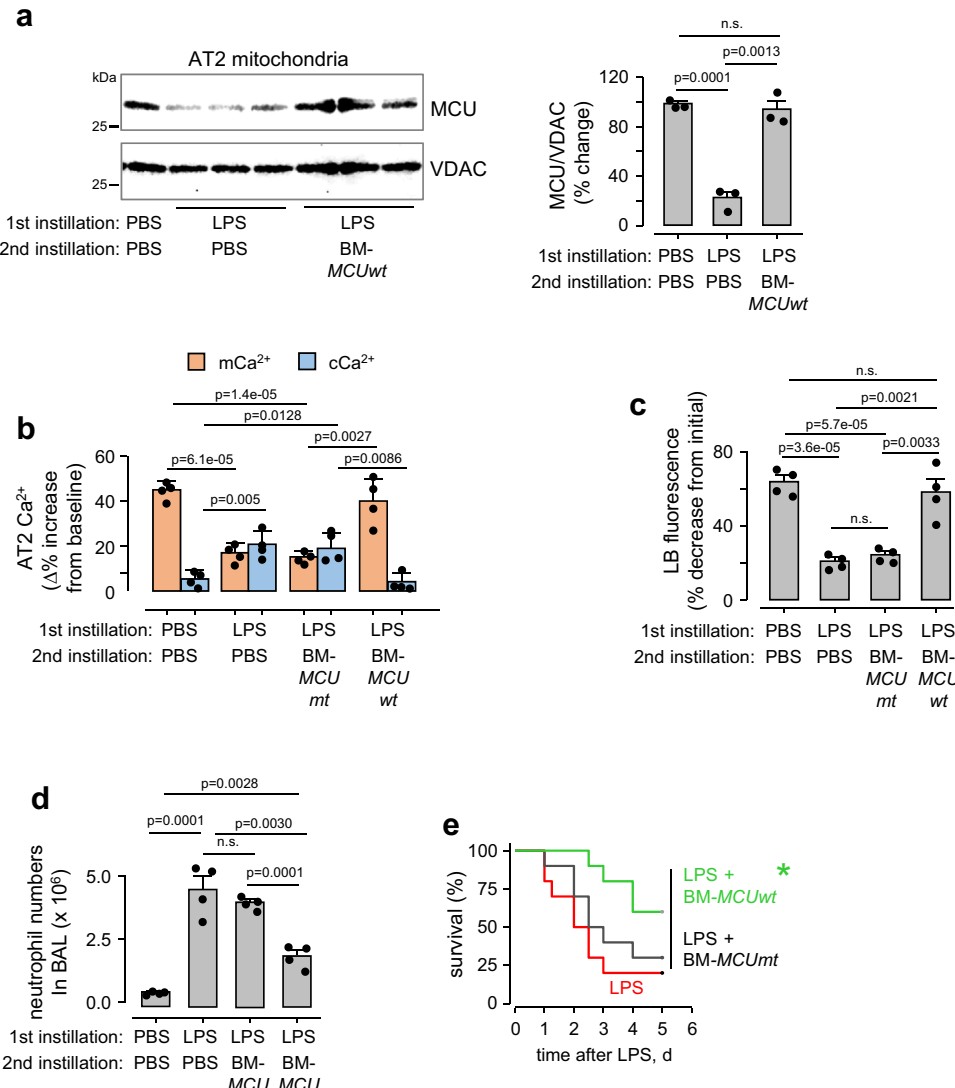

**Fig. 4 | BMSC expression of *pMCUwt* protects against LPS injury. a** MCU immunoblots in AT2 mitochondria derived from mice given indicated intranasal instillations. BMSCs expressing the wild-type MCU plasmid (*BM-MCUwt*) were intranasally instilled 4 h after nonlethal LPS instillations. Lungs were excised and AT2 isolated 24 h after LPS (1 mg/kg) instillations. Bars show densitometry. Bars: mean ± SEM. *n* = 3 lungs for each group. Groups were compared using one-way ANOVA with Bonferroni correction. *n.s.*, not significant. **b, c** Group data are for AT2 cytosolic (*cCa²⁺*) and mitochondrial (*mCa²⁺*) calcium (*b*) and surfactant secretion (*c*, loss of LTR fluorescence) responses following hyperinflation. Sequence of intranasal instillations is indicated. All determinations were carried out 24 h after the first instillation. *BM-MCUmt*, BMSCs expressing a mutant MCU. Bars: mean ± SEM.

*n* = 4 lungs each bar. Groups were compared using one-way ANOVA with Bonferroni correction. *n.s.*, not significant. *In c*, *p*-values are for two-tailed Student's *t*-test, *n.s.*, not significant. **d** Bars quantify the alveolar inflammatory response to indicated instillations as quantified by neutrophil counts in the bronchioalveolar lavage (BAL) obtained 24 h after the first instillation. Bars: mean ± SEM. *n* = 4 lungs each bar. Groups were compared using one-way ANOVA with Bonferroni correction. *n.s.*, not significant. **e** Mouse survival after instillations of lethal LPS (10 mg/kg) in *Swiss Webster* mice. The groups are: LPS alone (*red*), LPS followed 4 h later with instillation of BM-*MCUwt* (*green*) or BM-*MCUmt* (*black*). *N* = 10 each group, *\*p* = 0.038 versus LPS by Log-Rank test.

content. As we show here and have reported[21], LPS increases alveolar epithelial cCa²⁺ that could activate Ca²⁺-induced barrier-weakening mechanisms, such as calpain activation[48], or actin depolymerization[49]. Oxidant release by recruited leukocytes may damage the air-blood barrier causing hyperpermeability[50–52]. Pulmonary edema may have resulted from the combined effects of LPS-induced barrier-weakening mechanisms, and hyperfiltration effects due to surfactant loss. Hence, we propose that enhancement of endogenous surfactant secretion might be an alternative strategy for reversing ALI. In support, we show here that MCU replenishment through mitochondrial therapy in LPS-treated lungs reinstated surfactant secretion and promoted survival.

Mitochondrial H₂O₂ may play multiple mechanistic roles in ALI[53]. We reported that AE-derived H₂O₂ mediates paracrine

proinflammatory effects on the adjoining endothelium[4]. Here, the time course of MCU expression was unchanged from baseline for about 8 h post-LPS. Subsequently the expression progressively decreased by Day 1. In parallel, MCU function was intact early but not late after LPS exposure, while mitochondrial H₂O₂ progressively increased. AT2-specific catalase expression blocked the MCU loss, implicating mitochondrial H₂O₂ as a determinant of MCU expression. We conclude, increased production of mitochondrial H₂O₂ destabilized mitochondrial buffering, contributing to ALI lethality.

Increase of mitochondrial H₂O₂ in AT2 can occur from Complex 1 instability induced by binding of TRAF6 to ECSIT[44,45], or by Ca²⁺ communication across Cx43-containing GJs in the AE[22,42]. We affirmed these mechanisms, in that heterozygous ECSIT knockout, or AT2-specific

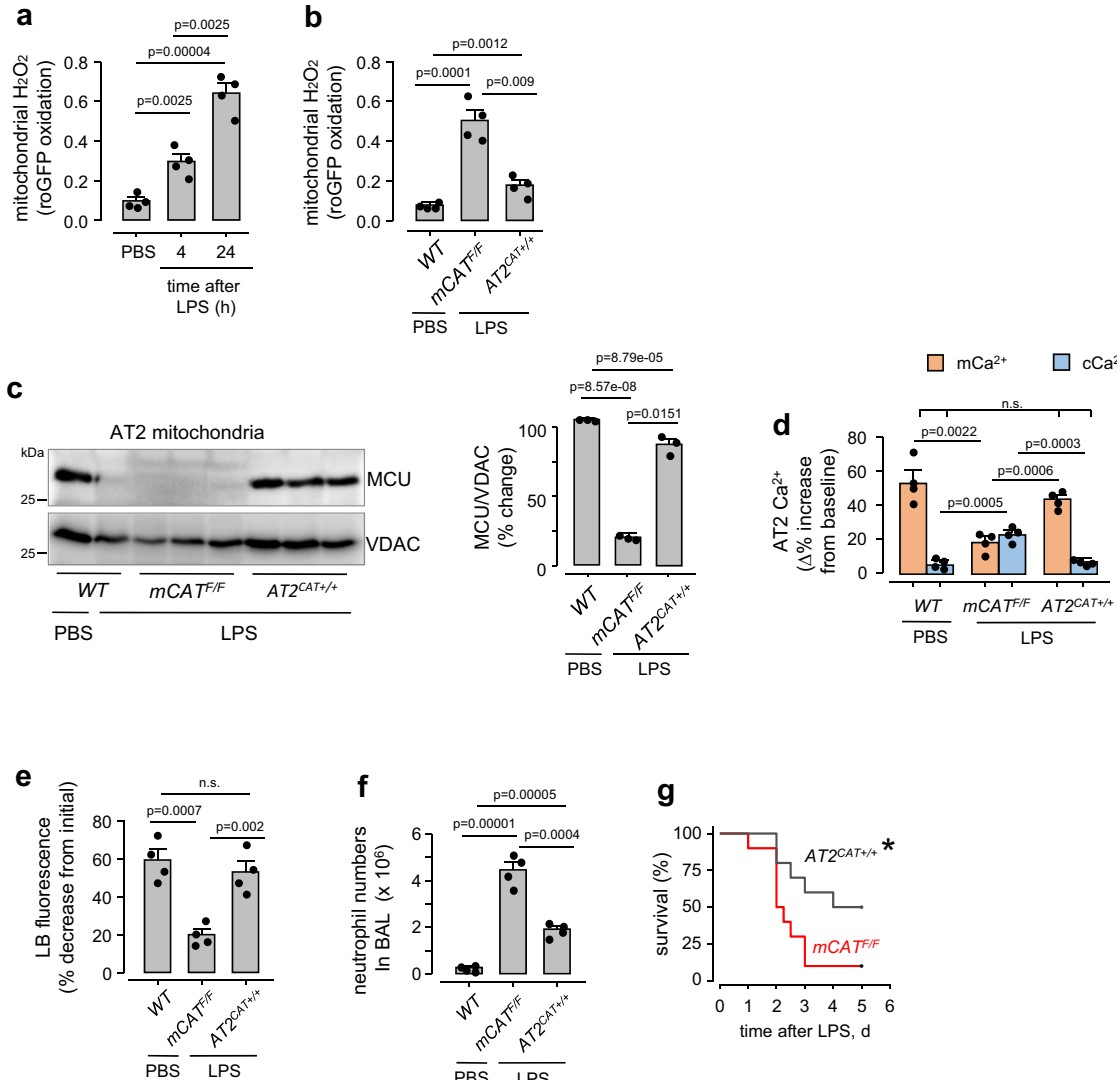

**Fig. 5 | Inhibition of mitochondrial $H_2O_2$ abrogates AT2 MCU depletion. a, b** Group data are for baseline AT2 mitochondrial $H_2O_2$ production following nonlethal LPS (1 mg/kg) instillations. $mCAT^{F/F}$, mice floxed for mitochondrial catalase ($mCAT$): $AT2^{CAT+/+}$, mice expressing mCAT in AT2. In **b**, determinations were made 24 h after indicated instillations. LPS was instilled at a nonlethal dose (1 mg/kg). Bars: mean ± SEM. $n = 4$ lungs for each group. Groups were compared using one-way ANOVA with Bonferroni correction. **c** MCU immunoblots and densitometry of AT2 mitochondria from lungs given intranasal PBS or nonlethal LPS (1 mg/kg). Lungs were excised and AT2 isolated 24 h after instillations. Bars: mean ± SEM. $n = 3$ lungs for each group. Groups were compared using one-way ANOVA with Bonferroni correction. **d**–**f** Group data show in situ determinations of AT2 cytosolic ($cCa^{2+}$) and mitochondrial ($mCa^{2+}$) calcium (**d**), surfactant secretion (**e**) and lung inflammation (**f**). All determinations were made 24 h after intranasal instillations. LPS was instilled at a nonlethal dose (1 mg/kg). Bars: mean ± SEM. $n = 4$ lungs each bar. Groups were compared using one-way ANOVA with Bonferroni correction. $n.s.$, not significant. **g** Kaplan-Meier plots are for mouse survival after instillations of ALI-inducing lethal LPS (50 mg/kg). $n = 10$ mice in each group, $*p = 0.027$ versus $mCAT^{F/F}$. $p$-value is calculated by Log-Rank test.

Cx43 knockout each markedly abrogated the LPS-induced MCU loss. These mechanisms may have operated in tandem, the $cCa^{2+}$-induced effect operating in the early phase of the immune response when mitochondrial buffering was present. In the later phase, as buffering failed, the ECSIT-induced effect may have dominated. Together, these mechanisms may have sustained $H_2O_2$ increase. $H_2O_2$ production activated Drp1, an effect that was inhibited by expression of mitochondrial catalase. Importantly, MCU loss was blocked in mice lacking *Drp1* in AT2. Taking these findings together, we conclude that $H_2O_2$-activation of Drp1 was the critical mechanism underlying the LPS-induced MCU loss. During immune challenge, MCU decrease may by negative feedback, block $mCa^{2+}$-induced $H_2O_2$ production.

Drp1 activation, resulting from exposure to viruses, bacteria and other inflammogens[54–57] is implicated in the mitochondrial fragmentation that precedes mitophagy, a process by which the cell eliminates dysfunctional mitochondria[36]. Here, LPS caused mitochondrial fragmentation as indicated by a loss of their polarized aggregated formations, and to absence of fusion-fission activity. However, contents of several proteins of the mitochondrial inner and outer membranes, as well as the mitochondrial membrane potential, a measure of mitochondrial fitness, were unchanged. Further, parkin localization to mitochondria, a marker of mitophagy initiation[26,40], was undetectable. Hence, we interpret that the fragmentation did not cause loss of AT2 mitochondria, and that the LPS-induced MCU loss was not due to Drp1-activated mitophagy of AT2. In unstressed lungs, AT2 mitochondria clustered with LBs, suggesting that the close LB-mitochondria proximity facilitated transfer of mitochondrial ATP to LBs to activate surfactant secretion. Diminished ATP transfer following mitochondrial disaggregation might account for inhibition of LPS-induced loss of surfactant secretion.

A limitation of our study is that the mechanisms underlying the ALI-induced MCU loss remain unclear. The loss was selective, since

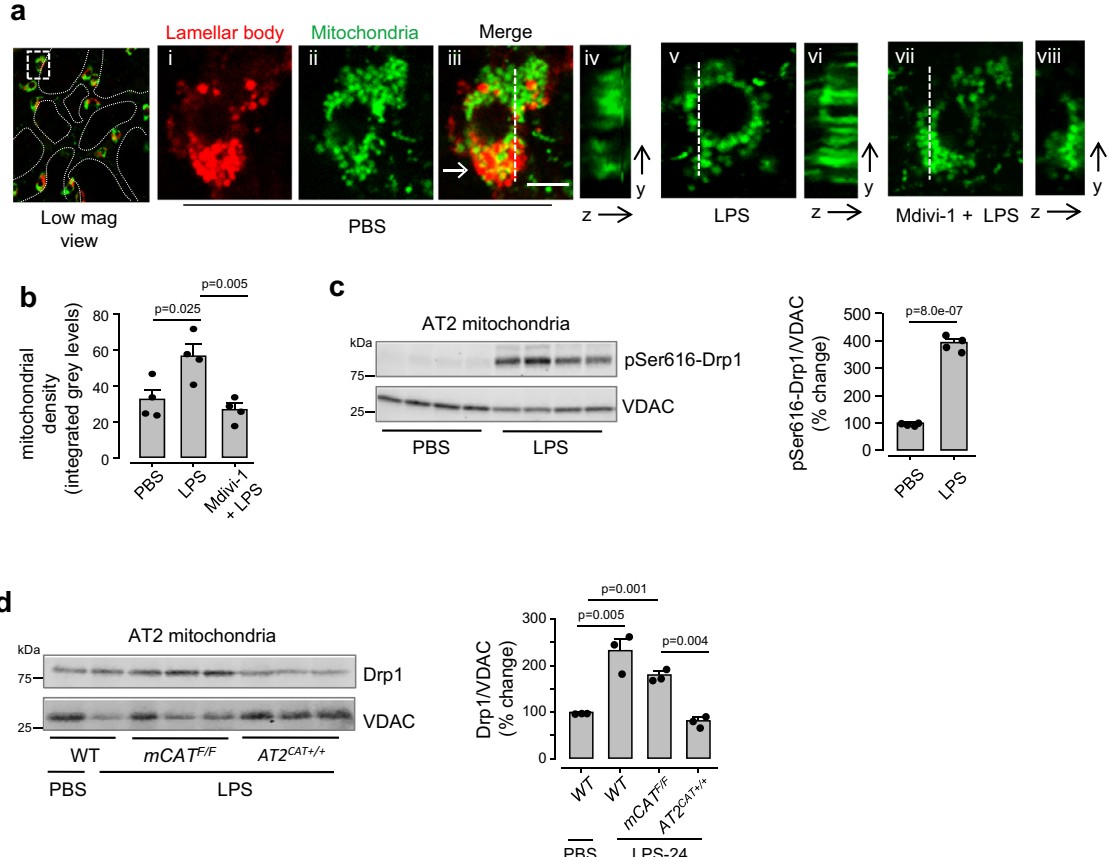

**Fig. 6 | LPS causes mitochondrial fragmentation. a**, **b** In mice expressing mitochondria-targeted dendra-2, we detected AT2 in terms of the lamellar body (LB) localizing dye, LTR. In the PBS-treated lung (*alveoli marked by dotted lines*), an AT2 is selected in the low magnification image (*rectangle*). Images *i-iii* show the AT2 at high magnification, displaying polarized mitochondrial aggregation. LBs co-mingle with mitochondria at the indicated site (*arrow*). Images *v* and *vii* display the channel for mitochondrial fluorescence in AT2 in a lung given intranasal instillation of nonlethal LPS (1 mg/kg, 24 h), and a lung that was given Mdivi-1 prior to the LPS treatment. Images *iv, vi,* and *viii* show distribution of mitochondrial fluorescence along the depth axes (*y-z*). Mitochondrial density was quantified in the depth axis at several sites of highest mitochondrial aggregation along the selected lines (*dashed lines*). Scale bar, 5 μm. Bars: mean ± SEM. *n* = 20 cells from 4 lungs each bar. Groups were compared using one-way ANOVA with Bonferroni correction. **c**, **d**, Phosphorylated Ser616-Drp1 (*c*) and Drp1 (*d*) immunoblots and densitometry are for freshly isolated AT2 mitochondria following indicated treatments. LPS was instilled at a nonlethal dose (1 mg/kg). *Drp1*, dynamin-related protein 1; *pSer616*, phosphorylated serine 616. Bars: mean ± SEM. *n* = 4 and 3 lungs each bar respectively, in *c* and *d*. In *c*, *p*-values are for two-tailed Student's *t*-test. In *d*, groups were compared using one-way ANOVA with Bonferroni correction.

several proteins of the mitochondrial outer and inner (IMM) membranes, the matrix, and the electron transport chain were not lost. Notably, in the post-LPS period, *MCU* mRNA was about 50% of control levels, indicating that *MCU* transcription was substantially active. This finding was further affirmed in that *MCU* transfection in the post-LPS period increased cytosolic MCU, but not mitochondrial MCU. One possibility is that LPS impaired mitochondrial import mechanisms. Since LPS caused mitochondrial redistribution, we speculate that mitochondrial import may have been impaired by loss of ER-mitochondrial tethering due to mitochondrial fragmentation[58,59], or Drp1 activation[59–61]. These proposed mechanisms need to be further understood in the ALI context.

Mitochondrial proteases known as ATPases associated with diverse cellular activities (AAA) may also have contributed to the LPS-induced MCU loss[62,63]. In this regard, attention has focused on the AAA proteases, AFG3L2 and SPG7. These proteases curate MCU function by degrading unassembled subunits of the MCU gatekeeper, EMRE[63,64]. Interestingly, knockouts of these proteases do not modify MCU expression[64,65], diminishing their possible role in the present MCU loss. Identification of new MCU-specific proteases might clarify the issue.

In conclusion, an important understanding that emerges relates to the role of MCU-dependent surfactant secretion in the pathophysiology of ALI. It is widely held that inflammatory cells, such as activated neutrophils that are rapidly recruited by the pathogen-challenged lung, cause barrier hyperpermeability and ALI[50,51]. However, lung infection in leukopenic patients can also cause ALI[66]. Hence, non-neutrophil mechanisms of ALI must also be considered, such as the MCU-surfactant mechanism we propose here. Further studies are required to better understand the role of the MCU mechanism in ALI under multiple predisposing conditions, including viral infections and malignant leukopenia.

## Methods

### Reagents, plasmids, primers, and antibodies
A list is provided in Supplementary Tables 1 and 2.

### Animals
Animal procedures were approved by the Institutional Animal Care and Use Committee of Vagelos College of Physicians and Surgeons at Columbia University. All animals were cared for according to the NIH guidelines for the care and use of laboratory animals. Mice were housed under a 12 h light/dark cycle with *ad libitum* access to water and food. All experimental animals were 5–10 weeks old and age- and sex-matched. Mice used are listed in Supplementary Table 1. Briefly, the mice used were: *Wild type: C57BL/6J, FVB-NJ* (The Jackson Laboratories); *Swiss Webster* (Taconic). *Cre recombinase expressing mice: SPC-*

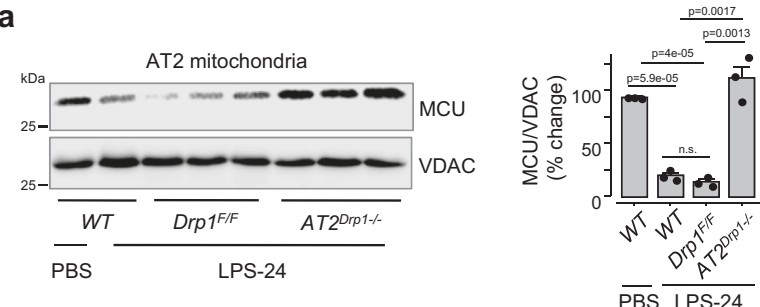

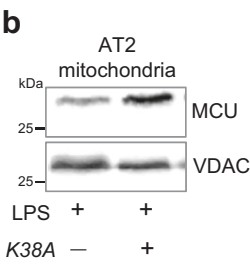

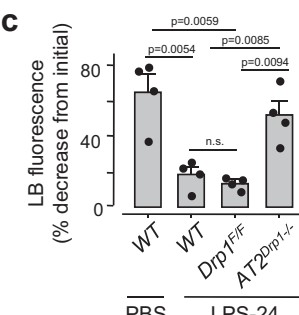

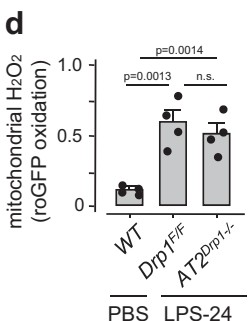

**Fig. 7 | LPS causes H₂O₂-induced Drp1 activation. a** MCU Immunoblots and densitometry are shown for AT2 mitochondria derived from lungs of mice given the indicated treatments. LPS was instilled at a nonlethal dose (1 mg/kg). *Drp1^F/F*, floxed mice for Drp1; *AT2^Drp1−/−*, mice lacking Drp1 in AT2. Lungs were excised and AT2 isolated 24 h after intranasal instillations. Bars: mean ± SEM. *n* = 3 lungs for each bar. Groups were compared using one-way ANOVA with Bonferroni correction. **b** MCU immunoblots in AT2 mitochondria derived from mice expressing an empty vector or a kinase-dead Drp1 (*K38A*). All animals were given intranasal LPS at a nonlethal dose and lungs excised 24 h after instillation. *n* = 3 lungs for each group. **c, d**, Group data show respectively, surfactant secretion (*c*) and mitochondrial H₂O₂ (*d*) following indicated treatments. All determinations were made 24 h after instillations. Bars: mean ± SEM. *n* = 4 lungs for each bar. Groups were compared using one-way ANOVA with Bonferroni correction.

---

*rtTa-tetO-Cre*, *SPC-ERT2-Cre*. Mice with floxed genes: *MCU^F/F*: *mCAT^F/F*; *Cx43^F/F*; *Drp1^F/F*. Other: *Dendra-2* expressing mice. To delete *MCU* in the alveolar epithelium (*rtTa^MCU−/−*), we crossed *MCU^F/F* mice[13] with *SPC-rtTa-Cre mice*[67]. The deletion was induced pre-partum through *SPC-Cre* induction by doxycycline treatment[21,23]. To delete MCU in AT2 cells (*ERT2^MCU−/−*), we crossed *MCU^F/F* mice with the *SPC-ERT2-Cre* mice, then induced Cre by post-partum tamoxifen treatment (75 mg/kg body weight, i.p. for 5d[68]). Controls (*ERT2Cre^inactv*) were mice given corn oil. To overexpress mitochondrial catalase (*mCAT*) in AT2 (*AT2^CAT+/+*), we crossed *SPC-rtTa-Cre* mice with *mCAT^F/F* mice[69], then activated the Cre post-partum by doxycycline treatment. To delete *Cx43* (*AT2^Cx43−/−*) and Drp1 (*AT2^Drp1−/−*) in AT2, we crossed, respectively, *Cx43^F/F* and *Drp1^F/F* mice with *SPC-rtTa-Cre*. To achieve AT2-specifc deletion, we induced the SPC-Cre post-partum[21,23].

**Acute Lung Injury (ALI)**
ALI was induced in anesthetized (ketamine-100 mg/kg and xylazine 5 mg/kg, i.p.) animals by airway instillation of LPS (*E.coli* 0111.B4) in sterile PBS. Control animals were instilled an equal volume of sterile PBS. Since multiple mouse strains were used, and since the lung injury-causing LPS dose varies between mouse strains, we gave different LPS doses to simulate sublethal, moderate and lethal ALI-inducing that caused mortality of 0, 30 and 80%, respectively, in control groups (Supplementary Table 3).

**Isolated, blood-perfused lungs**
Using our reported methods[18,20,21], lungs were excised from anesthetized mice, then perfused with autologous blood through cannulas in the pulmonary artery and left atrium. The blood was diluted in 4% dextran (70 kDa), 1% fetal bovine serum and buffer (150 mmol/l Na⁺, 5 mmol/l K⁺, 1.0 mmol/l Ca²⁺, 1 mmol/l Mg²⁺, and 20 mmol/l HEPES at pH 7.4). The perfusion flow rate was 0.5 ml/min at 37 °C at osmolarity of 300 mosM (Fiske Micro-Osmometer, Fiske® Associates, Norwood, MA), and hematocrit 10%. The lung was inflated through a tracheal cannula with a gas mixture (30% O₂, 6% CO₂, balance N₂). Vascular (artery and vein at 10 and 3 cmH₂O) and airway (5 cmH₂O) pressures were held constant during microscopy.

**Alveolar microinfusion and imaging**
To load the alveolar epithelium with fluorescent dyes or antibodies, we micropunctured single alveoli with glass micropipettes (tip diameter 3–5 µm) and microinfused -10 neighboring alveoli[70,71]. After the microinfusions, the free liquid in the alveolar lumen drained in seconds re-establishing air-filled alveoli[71]. This rapid clearance indicates that the micropuncture does not rupture the alveolar wall, and that the micropunctured membrane rapidly reseals as reported for other cells[72]. We selected non-micropunctured alveoli for imaging. To confirm that the fluorescence was intracellular, we microinfused alveoli for 10 min with trypan blue (0.01% w/v), which eliminates extracellular fluorescence[20]. In all experiments in which we infused multiple dyes, we confirmed absence of bleed-through between fluorescence emission channels. We imaged intact alveoli of live lungs with laser scanning microscopy (LSM 510 META, Zeiss and TCS SP8, Leica).

**Alveolar immunofluorescence**
We used our reported methods to detect intracellular immunofluorescence in live alveoli[18,21]. Briefly, we gave alveoli successive 20-minute microinfusions of 4% paraformaldehyde and 0.1% triton X-100. Then, we microinfused fluorescence-conjugated antibodies (40 ng/ml) for 10 min, followed by microinfusion of fluorescence-conjugated

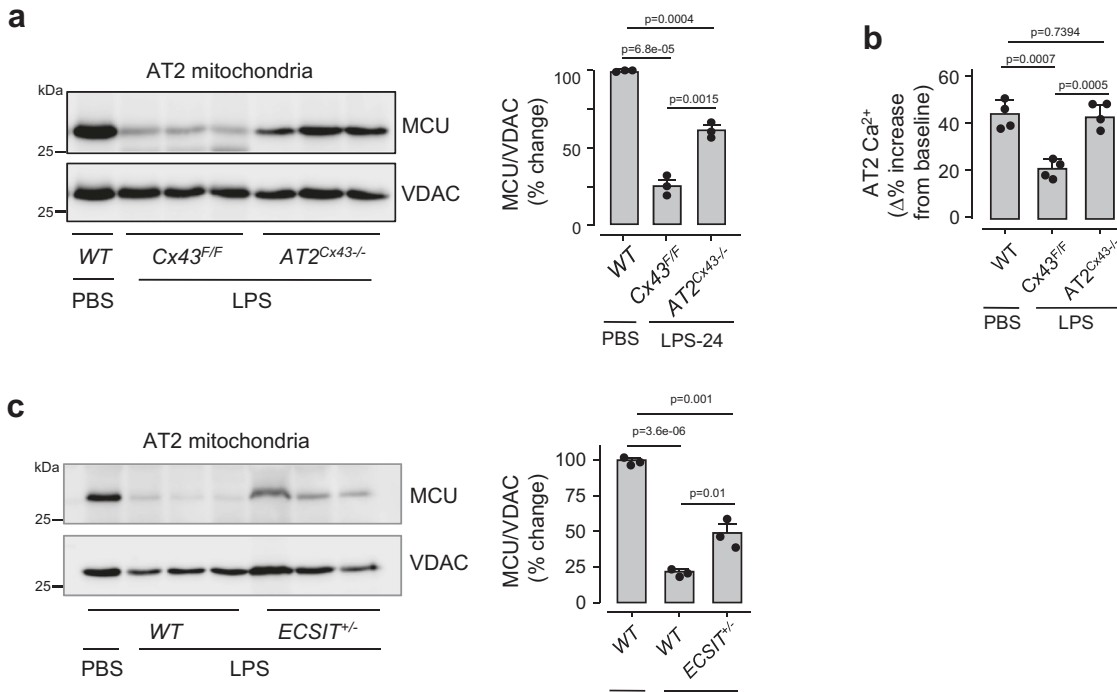

**Fig. 8 | Mechanisms of mitochondrial H₂O₂ generation. a** Immunoblots in AT2 mitochondria (*left*) and densitometric quantification of immunoblots (*right*) show MCU expression 24 h after indicated treatments. LPS was instilled at a nonlethal dose (1 mg/kg). *Cx43*, connexin 43; *Cx43^F/F^*, mice floxed for *Cx43*; *AT2^Cx43−/−^*, mice lacking *Cx43* in AT2. Bars: mean ± SEM. *n* = 3 lungs each bar. Groups were compared using one-way ANOVA with Bonferroni correction. **b** Determinations of mitochondrial Ca²⁺ responses to hyperinflation 24 h after indicated intranasal instillations. LPS was instilled at a nonlethal dose. Bars: mean ± SEM. *n* = 4 lungs each bar. Groups were compared using one-way ANOVA with Bonferroni correction. **c** Immunoblot (left), and densitometric determinations (right) from AT2 mitochondria. AT2 were isolated 24 h after instillations of either PBS or nonlethal LPS. *ECSIT*, evolutionarily conserved signaling intermediate in Toll pathway; *ECSIT^+/−^*, heterozygous knockout mice for ECSIT. Bars: mean ± SEM. *n* = 3 lungs each bar. Groups were compared using one-way ANOVA with Bonferroni correction.

secondary antibodies (40 ng/ml). To washout unbound fluorescence, we microinfused buffer for 10 min and commenced imaging after a further 10 min.

**Bacterial culture**

We cultured single colonies of *Pseudomonas aeruginosa* (strain K, provided by Dr. A. Prince, Columbia University, New York, NY, USA), overnight in 4 ml of Luria–Bertani (LB) medium at 37 °C and 250 rpm (Innova42, New Brunswick Scientific). *P. aeruginosa* were selected with Carbenicillin (300 μg/ml). On the day of the experiment, overnight cultures were diluted 1:100 in fresh LB and grown in a shaking incubator to optical density (OD) 0.5 at 600 nm (SPECTRAmax Plus, Molecular Devices).

**In vivo bacterial instillation**

For intranasal instillations in mice, 1 ml bacterial culture was centrifuged and resuspended in 10 ml sterile PBS. In anesthetized animals, we airway instilled 50 μl suspension to deliver "high" inoculum (1 × 10⁶ CFU) per mouse. For "low" inoculum instillation, we diluted (x10) the "high" inoculum suspension, then airway instilled 50 μl suspension to deliver 1 × 10⁵ CFU per mouse.

**Bone marrow-derived stromal cell (BMSC) isolation, purification and culture**

We isolated mouse BMSCs according to our reported methods[18]. Briefly, in anesthetized animals, bone lumens of excised femurs and tibia were infused with MSC growth medium to recover marrow. Three days after marrow plating on tissue culture flasks, the supernatant was rejected and adherent BMSCs were incubated (37 °C) under sub-confluent conditions to prevent cell differentiation. BMSCs from passages 5 to 12 were used in this study. In accordance with established criteria[73], BMSCs were characterized as we reported[18]. BMSCs were cultured in 5% CO₂ at 37 °C in DMEM containing 10% FBS and 1% of an antibiotic mixture.

**AT2 isolation by flow cytometry**

We isolated AT2 by our reported methods[18]. Briefly, isolated lungs were buffer perfused through vascular cannulas to clear blood, then we exposed the lungs to intratracheal dispase (0.2 U/ml, 2 ml, 45 min) at room temperature. The tissue was suspended in PBS and sieved and the sieved sample was then centrifuged (300 g x 5 min). The pellet was resuspended and incubated together with the AT2 localizing dye, lysotracker red (LTR)[42], the nuclear dye, Hoechst 33342, and fluorescence Allophycoyanin (APC)-conjugated Ab against the leukocyte antigen, CD45. The suspension was then subjected to cell sorting (Influx cell sorter) to recover AT2.

**Mitochondria isolation**

We used a commercially available isolation kit (Thermo Scientific)[20]. Briefly, we homogenized lungs and AT2 (Tissue Tearor; Biospec Products, Bartlesville, OK), on ice with buffers provided in the kit. The buffers were supplemented with protease and phosphatase inhibitors. We centrifuged the homogenate (800 *g* x 10 min) and collected the supernatants, which were further centrifuged at 17,000 *g* for 15 min at 4 °C. The pellets and the supernatants contained respectively, the mitochondrial and the cytosolic fractions. Purity of cell fractionation was determined by

immunoblotting for voltage-dependent anion channel (VDAC) in mitochondrial and cytosolic fractions.

## Immunoblotting/Immunoprecipitation
For immunoprecipitation and immunoblotting, mitochondrial fractions were lysed in 2% SDS[18]. For immunoprecipitation, lysates containing 2 mg total protein were precleared with appropriate control IgG for 30 min at 4 °C with 20 μl of prewashed protein A/G. Equal amounts (100 μg) of protein from lysates were separated by SDS-PAGE, electro-transferred onto nitrocellulose membrane overnight at 4 °C and blocked in Starting Block Blocking Buffer (Pierce) for 1 h then subjected to immunoblotting. Densitometry was performed using ImageJ software.

## RNA determination
RNA was extracted from isolated AT2 using the RNeasy Micro Kit. The purity of the RNA was assessed by absorbance at 260 and 280 nm using a Thermo Scientific NanoDrop spectrophotometer. Using RNA with a 260/280 ratio of >1.8, cDNA was synthesized using oligo (dT) and Superscript II. Quantitative RT-PCR was performed using a 7500 Real-Time PCR system (Applied Biosystems) and SYBR Green Master Mix.

## Lung transfection and knockdown
A list of plasmids and oligonucleotides used are listed in Supplementary Table 1. We purchased siRNA against MCU and RISP. For knockdown experiments, we intranasally instilled mice with 50 μg siRNA complexed with freshly extruded liposomes[18,20,21]. Stock solutions of plasmids (2.5 μg/μl) were similarly complexed with freshly extruded unilamellar liposomes (20 μg/μl, 100 nm pore size) in sterile PBS to a final concentration of 1 μg oligonucleotide/μl. Mice were intranasally instilled (50–75 μl) the nucleic acid-liposome mixture. Lungs from transfected animals were excised 48 h later. Protein expression or knockdown was confirmed by immunoblotting.

## Channelrhodopsin activation
To activate ChR2 in the alveolar epithelium, imaged alveolar fields were excited by mercury lamp illumination directed through an FITC (470 ± 10 nm) filter. The duration of field excitation was 10-seconds.

## Mitochondrial complex activity assay
To determine complex I-IV activities[74], purified mitochondria were resuspended in complexes I-IV activity assay buffers. Activity of complexes I and II were measured as the rate of decrease in absorbance at 600 nm. Whereas, activity of complexes III and IV were measured respectively, as the rate of increase and decrease in absorbance at 550 nm. For each assay, absorbance was measured at the given wavelength every 10 s for 30 min at 25 °C using a SpectraMax microplate spectrophotometer. Enzymatic activities were normalized to protein concentrations. Complex I activity was calculated as the difference between activities measured when samples were incubated with 2 μM Rotenone or ethanol.

## Extravascular lung water
We determined blood-free lung water content by our reported methods[75,76]. Briefly, we determined the wet and dry weights of excised lungs and quantified hemoglobin concentrations in lung homogenates to correct for blood water content in the wet/dry ratio.

## Lung ATP determination
We used our reported methods[18]. Briefly, excised lungs were immediately immersed in liquid N2. Lung samples (~10 mg) were subsequently subjected to colorimetric assays for determinations of ATP and protein content.

## Bronchioalveolar lavage (BAL) neutrophil count
BAL fluid was obtained following intratracheal instillations of 1 ml ice-cold sterile PBS. The BAL was centrifuged (400 g x 5 min at 4 °C), then the pellet resuspended in PBS supplemented with 1% BSA. To count neutrophils in the resuspended sample, we removed RBCs by exposing the sample to RBC lysis buffer. Then, we added fluorescent Ly-6G antibody to the sample, which we viewed by fluorescence microscopy in a Neubauer chamber (Olympus AX70).

## Quantification and statistical analysis
All major groups comprised a minimum of 3 mice each. Age and sex matched groups were randomly assigned. All data analysis was carried out by blinded protocol. Mean±SE was calculated on a per-lung basis for each group. Differences between groups were analyzed by the Student's $t$-test for two groups, and ANOVA with the Bonferroni correction for multiple groups. Survival comparisons were analyzed by the Logrank test. Significance was accepted at $P < 0.05$.

## Reporting summary
Further information on research design is available in the Nature Research Reporting Summary linked to this article.

## Data availability
Source Data for all figures are provided. All data supporting the findings described in this manuscript are available in the article and in the Supplementary Information and from the corresponding author upon reasonable request. Source data are provided with this paper.

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

## Acknowledgements

This work was supported by NIH grants HL36024, HL57556, and HL122730 to J.B., and a Parker B. Francis Fellowship and an American Heart Association (AHA) Grant-in-Aid to M.N.I. L.M. was supoted by an AHA Postdoctoral Fellowship (ID: 902655). Some studies were carried out in the Columbia Center for Translational Immunology Flow Cytometry Core, supported in part by NIH award S10OD020056. Michelle Wei assisted with genotyping. The PhAM floxed:E2a-Cre and ECSIT[+/-] mice were gift of Drs. Hans Snoeck and Sankar Ghosh (Columbia University), respectively. Purified human SPB was a gift of Dr. Timothy Weaver (University of Cincinnati).

## Author contributions

M.N.I. planned, carried out and analyzed all experiments. G.A.G. contributed to alveolar type 2 cell isolation and imaging experiments. L. L., E.M. and L.M. contributed to alveolar hyperinflation experiments. S.Q. contributed to experiments with *Pseudomonas aeruginosa*. M.A. and E.O-A. performed mitochondrial complex activity determinations. S.D. carried out immunoblotting. S.B. contributed to the plan. J.B. designed the overall project. All authors contributed to the writing.

## Competing interests

The authors declare no competing interests.
