## [Peer Review File · Nature Communications]

The Mitochondrial Calcium Uniporter of Pulmonary Type 2 Cells Determines Severity of Acute Lung InjuryREVIEWER COMMENTS

Reviewer #1 (Remarks to the Author):

In the report by Islam and colleagues, the authors provide evidence that LPS challenge results in the loss of MCU expression in AT2 cells of the lung. Through genetic and biochemical approaches they implicate the generation of hydrogen peroxide as a mediator of this MCU loss. They also implicate Drp1 activation, Cx43-containing gap junctions, and the Complex I assembly factor, ECSIT. Finally, the authors show that restoration of MCU reduces LPS-induced mortality. Overall, the results are of interest but there remains a major mechanistic gap that would strengthen the manuscript if addressed. In particular, the author's should attempt to further delineate by what means does the decline in MCU expression occur? While there is a modest fall in transcription, it would appear this represents a post-translational event. Given its inner mitochondrial membrane localization, I would assume that one of the resident mitochondrial proteases is involved. Some effort should be invested in delineating whether this is or isn't true. Furthermore, it would be nice to know whether hydrogen peroxide is sufficient for these effects and whether any direct modification to MCU occurs before it is subject to degradation. In addition, are other components of the uniporter (e.g. EMRE, MICU1) also subjected to this LPS-induced decline?

Reviewer #2 (Remarks to the Author):

This mechanistic manuscript advances our understanding of the role of calcium flux and mitochondrial dysfunction in alveolar epithelial cells during inflammatory lung injury. The data suggest that LPS induces downregulation of MCU (via mitochondrial H₂O₂ and Cx43 and ECSIT), which results in impaired mitochondrial Ca⁺⁺ influx, leading to increased cytoplasmic Ca⁺⁺, impairing ATP-dependent surfactant secretion, resulting in mortality. Mortality can be rescued by MCU transfection or delivery via BMSCs. There are however several concerns, both methodological and conceptual.

Major

1. The Cre driver used here has limitations and therefore has been widely replaced by the SPCCreERT2 driver. It is unclear whether the proper controls have been used to rule out adverse effects of the rtTa;tetOCre system. Standard mouse nomenclature should be used throughout. Specifically, in Fig. 1B, please specify the exact genotypes of the mice. Do both the MCU^{f/f} mice and AE MCU⁻ have the rtTA and tetOCre? Cre toxicity can occur, and the rtTA transgene in lung epithelial cells has been noted to cause emphysema (PMID 16415250), presumably due to epithelial cell dysfunction. Therefore, all the proper controls should be used to confirm that it is MCU deletion, not the Cre or rtTA that are causing AT2 cell dysfunction in this context. Better yet, SftpcCreERT2 mouse is the state-of-the-art AT2 cre driver that avoids many of the issues with the rtTA;tetOCre system.
2. The causal link between impaired mitochondrial Ca⁺⁺ buffering, surfactant secretion, barrier integrity, and mortality has not been demonstrated. First, as I understand it, it is not the change in mitochondrial Ca⁺⁺ per se that is thought to be important for surfactant secretion, but rather its ability to buffer cytoplasmic Ca⁺⁺. Therefore, the effect of LPS in the presence and absence of MCU on cytoplasmic Ca⁺⁺ should be shown in all figures. Second, the authors show that MCU KO increases mortality, but have not shown that MCU KO increases mortality by impairing surfactant secretion. There is no demonstration that rescuing surfactant secretion in the MCU KO prevents mortality. Therefore, it seems plausible that MCU KO increases mortality by preventing Ca⁺⁺ influx, thus causing a global mitochondrial dysfunction beyond simply an effect on surfactant secretion. The MCU-dependent mortality and weight loss observed are likely due to more than simply the effect on surfactant secretion. The authors imply that impaired surfactant secretion impairs barrier integrity but no permeability data are shown and it is unclear how impaired surfactant secretion would directly compromise

barrier function.

3. The authors need to determine whether the mice even die from respiratory failure. The authors demonstrate in some figures an effect on neutrophil influx but the causal role of neutrophils on barrier integrity and systemic effects such as weight loss and mortality have not been considered. Neutrophil depletion is protective in the LPS model (Rittirsch D, et. al. J Immunol 2008), so an AT2 autonomous effect of MCU loss on barrier permeability and mortality is unlikely. Remember that the 95-98% of the alveolar barrier is comprised of AT1 cells, so if the authors are postulating a role of barrier permeability, AT1 cells must be considered.
4. Similarly, the catalase transgenic mice do not increase mitochondrial H₂O₂ and do not lose MCU, suggesting that mitochondrial H₂O₂ causes MCU loss, but the effect of catalase on mortality may be related to effects of H₂O₂ other than the effect on MCU and surfactant secretion. The fact that catalase transgenic mice have less inflammation (Fig. 3f) suggests that there are more broad consequences of AT2 H₂O₂ than simply its effect on MCU levels (unless the authors propose that impaired surfactant secretion is the mechanism for enhanced neutrophil influx but then this must be shown also).
5. In sublethal LPS, the MCU levels recover and the mice survive. In lethal doses of LPS, the MCU levels do not recover, and the mice die (Fig. 2F). If the lethal dose of LPS causes complete loss of MCU (Fig. 2F), it's unclear why knocking out MCU in the lethal dose of LPS would increase mortality. KO cannot decrease MCU levels below zero. This raises further concern that Cre or rtTA toxicity in the MCU KO mice, rather than loss of MCU, is impairing AT2 function.
6. Use of the term ARDS is incorrect, it is a model of ACUTE LUNG INJURY from LPS
7. The study would be strengthened by lung water and BAL protein at key time points.
8. Overall, the basic hypothesis is supported with the imaging methods but the conclusion that the "failed surfactant secretion" causes the higher mortality seems a little limited, unless the loss of the MCU impairs other epithelial and lung functions that have not been measured or captured, as explained above.
9. In other words, why did the mice die? Was it from lack of surfactant as in neonatal respiratory distress syndrome in premature infants or was it from pulmonary edema and lung injury?
10. The add on Pseudomonas experiments have some value, but seem limited, even though they add some clinical relevance. What happened to these mice in terms of bacterial counts in the lung and the spleen for example? Were type 1 cells affected in their loss and gain of function studies?
11. Discussion lacks a proper self-critique.

Minor

12. In Fig. 2, the authors demonstrate that LPS causes decreased mCa⁺⁺, decreased surfactant secretion, and decreased MCU. To confirm that the LPS-induced impairment in surfactant secretion is due to decreased MCU, they should rescue the effect with MCU stabilization/overexpression. Fig. 7 should be moved up, after Fig. 2.
13. In the first sentence of the Results section, the model system should be described briefly.
14. Please define mCa²⁺ and cCa²⁺ at first mention.
15. The statement in the text that BAL leukocyte counts recover to baseline by day 3 of LPS (Extended data fig. 2a) seems wrong because the figure shows that counts don't return to baseline until day 5. Also, the authors suggest that extended data fig. 2b shows the abrogation of surfactant secretion in the LPS model, but those data are shown in main Fig. 2.
16. Please ensure that the y-axis in the surfactant secretion figures is correct and explain a little more. As I understood the explanation, the loss of fluorescence indicates secretion. However, if the y-axis is % initial fluorescence, the lower the bar seems to indicate more secretion, not less. So it is confusing to me why MCU inhibition/KO would result in more loss of fluorescence, which seems to indicate more secretion. It would also be helpful to show a pre-stretch/pre-LPS surfactant secretion level.
17. The abstract could be rewritten to highlight more of the mechanistic details in the paper.

18. All the statements causally link loss of barrier function to impaired surfactant secretion should be reworded or explained since the direct causal link between surfactant and barrier permeability is not immediately clear.

19. LPS doses should be included on the figures. Otherwise, it's unclear why LPS sometimes causes 100% mortality in WT mice (Fig. 7F) and sometimes does not (Fig. 2H).

Point-by-point response to Reviewers' comments

We thank the Reviewers for their critical and positive evaluations of our manuscript. We revised the paper extensively. Our revisions are marked in red font in the manuscript. In our response below, we refer to the revisions by line number.

Reviewer 1:

In the report by Islam and colleagues, the authors provide evidence that LPS challenge results in the loss of MCU expression in AT2 cells of the lung. Through genetic and biochemical approaches, they implicate the generation of hydrogen peroxide as a mediator of this MCU loss. They also implicate Drp1 activation, Cx43-containing gap junctions, and the Complex I assembly factor, ECSIT. Finally, the authors show that restoration of MCU reduces LPS-induced mortality. Overall, the results are of interest but there remains a major mechanistic gap that would strengthen the manuscript if addressed. In particular, the author's should attempt to further delineate by what means does the decline in MCU expression occur? While there is a modest fall in transcription, it would appear this represents a post-translational event. Given its inner mitochondrial membrane localization, I would assume that one of the resident mitochondrial proteases is involved. Some effort should be invested in delineating whether this is or isn't true. Furthermore, it would be nice to know whether hydrogen peroxide is sufficient for these effects and whether any direct modification to MCU occurs before it is subject to degradation. In addition, are other components of the uniporter (e.g. EMRE, MICU1) also subjected to this LPS-induced decline?

Thank you for the kind comments. While we agree with that there is a mechanistic gap, we respectfully point out that our main goal was to report the novelty of the MCU-surfactant mechanism as a mortality-causing factor in ALI. We note that the Reviewer appears to have no issues with this aspect of our paper. As regards the MCU loss, we agree that an LPS-induced post-translational mechanism might underlie the effect. To test this possibility, we carried out new experiments in which we assayed the extent to which episomally transcribed MCU reaches the mitochondria. Under control conditions this transport was robust, since MCU transfection enriched mitochondrial MCU (Extended Data Figure 3c). However in marked contrast, 12h after LPS, although MCU was expressed in the cytosol, indicating presence of episomal transcription, there was no corresponding increase of MCU in mitochondria (new Figure 3h). These findings suggest that LPS caused impairment of mitochondrial import of the MCU from cytosolic transcription sites, accounting for the LPS-induced loss of mitochondrial MCU. We recognize that further studies are required to identify the possible import defect. However, since these studies are likely to be extensive, we are pursuing them as a separate project. We thank the Reviewer for drawing our attention to this issue, which we address in revised text in Results (Lines 180-185) and Discussion (Lines 334-342).

We took note of the Reviewer's suggestion regarding proteases. The difficulty is that a specific MCU-targeting protease remains unidentified. However, the issue is important, and we have included a discussion of proteases (Lines 343-348). We thank the Reviewer for the protease question, which we expect to pursue in a future project.

To fully address the effects of LPS on other uniporter components, we need a clearer understanding of the MCU loss mechanism. Thus, to the extent that the MCU loss is a mitochondrial import issue, as we discuss above, a primary modification of the MCU would probably not be required. Instead the modification is likely to involve components of the mitochondrial import machinery that brings in the MCU from the cytosol. Since LPS caused mitochondrial redistribution (Figure 6a), mitochondrial import may have been impaired by loss of ER-mitochondrial tethering, by the process of mitochondrial fragmentation, or by Drp1 activation,

as proposed by the papers we cite in the revised text (Lines 339-342). In our new project we hope to evaluate these issues in relation to import of the MCU as well as the MCU-associated proteins EMRE, and MICU1. We thank the Reviewer for these comments that will importantly shape our future research. However, we respectfully point out that we did not include these evaluations in the revision as that would greatly exceed present scope.

Reviewer 2:

We thank the reviewer for encouraging and thoughtful comments.

Response to Major Comments

#1 The Cre driver used here has limitations and therefore has been widely replaced by the SPCCreERT2 driver. It is unclear whether the proper controls have been used to rule out adverse effects of the rtTa;tetOCre system. Standard mouse nomenclature should be used throughout. Specifically, in Fig. 1B, please specify the exact genotypes of the mice. Do both the MCU^{fl/fl} mice and AE MCU⁻ have the rtTA and tetOCre? Cre toxicity can occur, and the rtTA transgene in lung epithelial cells has been noted to cause emphysema (PMID 16415250), presumably due to epithelial cell dysfunction. Therefore, all the proper controls should be used to confirm that it is MCU deletion, not the Cre or rtTA that are causing AT2 cell dysfunction in this context. Better yet, SftpcCreERT2 mouse is the state-of-the-art AT2 cre driver that avoids many of the issues with the rtTa;tetOCre system.

1.1. In response to the excellent suggestion regarding the Cre driver, we repeated key findings in the SPC-CreERT2 mouse in which the MCU gene was deleted in an AT2-specific manner. The new data show that MCU deletion (i) blocks inflation-induced mitochondrial Ca²⁺ buffering in AT2 cells; (ii) blocks surfactant secretion; and (iii) causes severe LPS-induced lethality that is reversed by MCU add back. These data are included in the following new figures: Figures 1d,e and Figures 3a,b. We conclude that our key findings could be replicated by two different drivers of Cre recombination, ruling out non-specific effects of Cre drivers. We revised the text to address the new studies with the SPC-ERT2-Cre mice (Lines 110-114, 161-169, 188-194, 276-278, 381-384, 417-428 and 444-455).

1.2. The reviewer raises important issues regarding our use of the rtTA:tetOCre system. In studies by Sisson et al¹, the rtTA transgene when driven by the promoter for the gene, CCSP, which is expressed in Club cells and in epithelial cells in the small and large airways, induced emphysema like changes. However, when driven by the SPC promoter, which is expressed in AT2 cells, the rtTA transgene did not cause significant adverse effects. Thus, per Sisson et al's findings, rtTA expression is not expected to majorly affect alveolar epithelial function. Nevertheless, as we point out in revised text (Lines 97-98), we compared alveoli of equal diameters to rule out emphysema as a confounding factor, and we obtained control data in rtTa-tetO-Cre expressing mice which were not exposed to doxycycline to rule out possible negative effects of transgene expression. In new experiments we show that major outcomes, such as mitochondrial buffering and surfactant secretion were similar in lungs expressing non-activated rtTA, or the floxed MCU allele (new Figures 1b,c). Therefore, no adverse effects were evident due to expression of the rtTA:tetOCre system.

1.3. As recommended, we corrected the manuscript to include standard mouse nomenclature.

#2. The causal link between impaired mitochondrial Ca⁺⁺ buffering, surfactant secretion, barrier integrity, and mortality has not been demonstrated. First, as I understand it, it is not the change in mitochondrial Ca⁺⁺ per se that is thought to be important for surfactant secretion, but rather its ability to buffer cytoplasmic Ca⁺⁺. Therefore, the effect of LPS in the presence and absence of MCU on

cytoplasmic Ca⁺⁺ should be shown in all figures. Second, the authors show that MCU KO increases mortality, but have not shown that MCU KO increases mortality by impairing surfactant secretion. There is no demonstration that rescuing surfactant secretion in the MCU KO prevents mortality. Therefore, it seems plausible that MCU KO increases mortality by preventing Ca⁺⁺ influx, thus causing a global mitochondrial dysfunction beyond simply an effect on surfactant secretion. The MCU-dependent mortality and weight loss observed are likely due to more than simply the effect on surfactant secretion. The authors imply that impaired surfactant secretion impairs barrier integrity but no permeability data are shown and it is unclear how impaired surfactant secretion would directly compromise barrier function.

2.1. We appreciate this thoughtful comment regarding the causal link. We extensively revised the text to indicate the link between mitochondrial buffering and surfactant secretion (Lines 57-62), the link between surfactant deficiency and mortality (Lines 186-194, 270-275, and 283-294), and the link between edema and mortality (Lines 295-303, and Figures 3b and 3i). Briefly, MCU deletion or LPS-induced MCU depletion, impaired mitochondrial buffering, resulting in surfactant deficiency. The resulting edema could be blocked by replenishing MCU, which restored buffering, or by pre-LPS surfactant replenishment. Our new experiments provide definitive evidence that pre-existing surfactant deficiency causes severe pulmonary edema in adult lungs.

2.2. As requested by the Reviewer, the effects of LPS on the cytosolic Ca²⁺ are now shown in Figures 1b, 2b, 3d, 4b, 5d and Extended Data Figure 1e.

2.3. Regarding the comment on the rescuing effect of surfactant, we include new evidence that surfactant replenishment in the MCU KO mice prevents mortality. Please see also Response #2.1 regarding our discussion of the causal link between surfactant secretion and mortality.

2.4. Regarding the comment on global mitochondrial dysfunction as an alternative to surfactant deficiency, to the extent we did not detect loss of major complexes of the mitochondrial electron transport chain, or of the mitochondrial structural proteins, TOM20, VDAC and HSP60, following LPS treatment, we may rule out global mitochondrial dysfunction as a factor in these responses. Further, as we discuss in Response #2.1, the survival protection following surfactant pre-treatment indicates that surfactant deficiency caused the mortality.

2.5. Regarding the comment relating impaired surfactant secretion and barrier function, we included new text to discuss the proposal that the interstitial pressure drops in surfactant deficiency (Clements hypothesis), causing hyperfiltration due to imbalance of Starling forces. This discussion is included in revised text (Lines 186-194, and 282-293).

#3. The authors need to determine whether the mice even die from respiratory failure. The authors demonstrate in some figures an effect on neutrophil influx but the causal role of neutrophils on barrier integrity and systemic effects such as weight loss and mortality have not been considered. Neutrophil depletion is protective in the LPS model (Rittirsch D, et. al. J Immunol 2008), so an AT2 autonomous effect of MCU loss on barrier permeability and mortality is unlikely. Remember that the 95-98% of the alveolar barrier is comprised of AT1 cells, so if the authors are postulating a role of barrier permeability, AT1 cells must be considered.

3.1. Regarding the comment on the cause of mouse mortality, we show in new Figure 3i, AT2-specific loss of MCU caused severe pulmonary edema, as evident in the large increase of lung water, indicating death was caused by respiratory failure. The addback studies (Figure 3b), indicate replenishing MCU in MCU-deficient AT2 cells protected against mortality. Hence, the respiratory failure was due to pulmonary edema resulting from loss of AT2 MCU.

3.2. The causal role of neutrophils in ALI is well documented in the report cited by the Reviewer, as well as in other reports that we now cite. However, as we discuss in revised text (Lines 351-356),

ALI can also occur in leukopenia. Hence, non-neutrophil mechanisms, such as the present MCU-surfactant mechanism, require consideration.

We did not re-address the neutrophil hypothesis through the suggested neutrophil depletion studies, as the studies would extend scope. However, the Reviewer's point is well taken, that the hyperpermeability arm of the LPS response needs to be better understood. We suggest that the hyperpermeability effect arose from LPS-induced effects on alveolar Ca^{2+} . To the extent buffering impairment increased cCa^{2+} , loss of MCU – the cause of buffering impairment – may have contributed to the hyperpermeability. We noted in revised text that these issues need further study (Lines 295-303).

3.3. Regarding the comment on AT1 cells, we consider that only AT2 cells secrete surfactant, the secretion covers the entire alveolar surface, including AT1 cells². The hyperfiltration effects of surfactant loss are likely to be most pronounced at all alveolar curvatures, irrespective of whether the curvatures contain AT1 or AT2 cells.

#4. Similarly, the catalase transgenic mice do not increase mitochondrial H_2O_2 and do not lose MCU, suggesting that mitochondrial H_2O_2 causes MCU loss, but the effect of catalase on mortality may be related to effects of H_2O_2 other than the effect on MCU and surfactant secretion. The fact that catalase transgenic mice have less inflammation (Fig. 3f) suggests that there are more broad consequences of AT2 H_2O_2 than simply its effect on MCU levels (unless the authors propose that impaired surfactant secretion is the mechanism for enhanced neutrophil influx but then this must be shown also).

The reviewer raises an interesting point. As we have reported³, alveolar catalase can diminish endothelial proinflammatory responses by hydrolyzing H_2O_2 that promotes neutrophil influx. However, as we show here, an additional negative effect of H_2O_2 is that it also abrogates mitochondrial buffering, hence surfactant secretion. Thus a dual barrier challenge is established – neutrophil influx-induced barrier damage, and the hyperfiltration effect of surfactant loss. Together these mechanisms caused major pulmonary edema in the AT2 specific MCU-deficient mice. We thank the Reviewer for raising this point, which we discuss in revised text (Lines 304-311, and 350-356).

#5. In sublethal LPS, the MCU levels recover and the mice survive. In lethal doses of LPS, the MCU levels do not recover, and the mice die (Fig. 2F). If the lethal dose of LPS causes complete loss of MCU (Fig. 2F), it's unclear why knocking out MCU in the lethal dose of LPS would increase mortality. KO cannot decrease MCU levels below zero. This raises further concern that Cre or rtTA toxicity in the MCU KO mice, rather than loss of MCU, is impairing AT2 function.

5.1. We regret the confusion regarding the extent of MCU loss after LPS. We revised the text to clarify that LPS repressed but did not cause complete loss of MCU (Lines 124-127). Moreover, substantial evidence for transcription was present, indicating MCU expression occurred, if at levels lower than control. The physiological outcomes, namely buffering and surfactant secretion were also repressed but not wholly blocked. When we knocked out *MCU*, thereby completely abolishing MCU expression, the mortality was greater. Thus, as we show in Extended Data Figure 3a, the same dose of LPS caused greater mortality in *MCU* knockout mice than in control, namely in *MCU* floxed mice. Further, we include new data showing severer mortality in mice with AT2-specific *MCU* knockout than in control mice in which Cre was not activated (Figure 3b).

5.2. Regarding the comment on Cre toxicity, our new data show that the higher mortality was also obtained in the ERT2-Cre mice, ruling out rtTA toxicity. Please see also Responses #1.1 and #1.2 in which we discuss the Cre models.

#6. Use of the term ARDS is incorrect, it is a model of ACUTE LUNG INJURY from LPS. We agree. This change has been made throughout.

#7. *The study would be strengthened by lung water and BAL protein at key time points.*

Data from new experiments show lung water and BAL protein data (new Figures 3i and 3j). The text was revised (Lines 188-191).

#8. *Overall, the basic hypothesis is supported with the imaging methods but the conclusion that the “failed surfactant secretion” causes the higher mortality seems a little limited, unless the loss of the MCU impairs other epithelial and lung functions that have not been measured or captured, as explained above.*
Please see response #2.1 in which we discuss the critical role of surfactant.

#9. *In other words, why did the mice die? Was it from lack of surfactant as in neonatal respiratory distress syndrome in premature infants or was it from pulmonary edema and lung injury?*

They died from severe pulmonary edema as we now clarify (Lines 186-194) and new Figures 3b and 3i.

#10. *The add on Pseudomonas experiments have some value, but seem limited, even though they add some clinical relevance. What happened to these mice in terms of bacterial counts in the lung and the spleen for example? Were type 1 cells affected in their loss and gain of function studies?*

Regrettably, we do not have these data. We restricted our studies to AT2 as our focus was on surfactant secretion. The question regarding possible responses in AT1 is an interesting and will be considered in a separate project.

#11. *Discussion lacks a proper self-critique.*

Thank you for pointing this out. The Discussion is revised to address the limitations of the study (lines 335-343).

Response to Minor Comments

#12. *In Fig. 2, the authors demonstrate that LPS causes decreased mCa^{++} , decreased surfactant secretion, and decreased MCU. To confirm that the LPS-induced impairment in surfactant secretion is due to decreased MCU, they should rescue the effect with MCU stabilization/overexpression. Fig. 7 should be moved up, after Fig. 2.*

MCU stabilization/overexpression did reinstate mitochondrial Ca^{2+} entry, rescue surfactant secretion, and reduce mortality. As recommended the data from old Fig. 7 are now included as new Fig. 3.

#13. *In the first sentence of the Results section, the model system should be described briefly.*

This is done (Line 83).

#14. *Please define mCa^{2+} and cCa^{2+} at first mention.*

This is done (Lines 58-59).

#15. *The statement in the text that BAL leukocyte counts recover to baseline by day 3 of LPS (Extended data fig. 2a) seems wrong because the figure shows that counts don't return to baseline until day 5. Also, the authors suggest that extended data fig. 2b shows the abrogation of surfactant secretion in the LPS model, but those data are shown in main Fig. 2.*

We apologize for the errors. As correctly noted by the Reviewer, we revised the text to clarify that the BAL leukocyte levels return to baseline by Day 5 (Line 120), and that the surfactant data should be in main Figure 2a, as we have now positioned (Line 121). We thank the Reviewer for noting these textual errors.

#16. Please ensure that the y-axis in the surfactant secretion figures is correct and explain a little more. As I understood the explanation, the loss of fluorescence indicates secretion. However, if the y-axis is % initial fluorescence, the lower the bar seems to indicate more secretion, not less. So it is confusing to me why MCU inhibition/KO would result in more loss of fluorescence, which seems to indicate more secretion. It would also be helpful to show a pre-stretch/pre-LPS surfactant secretion level.

16.1. Regarding the comment on the y-axis label, the Reviewer is correct that decrease of lamellar body (LB) fluorescence from baseline denotes surfactant secretion. Accordingly, we corrected the y-axis legends for the surfactant figures to clarify that the bars represent % decrease of LB fluorescence from initial. Thus in Figure 1c, in the control groups ('MCU^{F/F}' and 'Cre-inactive'), the fluorescence decrease was ~60% of initial, calculated as the difference of grey levels between initial and the 30 min value after response initiation, divided by initial. The 60% decrease denotes surfactant secretion. In the knockout, the % decrease was negligible, hence surfactant secretion was negligible. We clarified this in revised legend for Figure 1c (Lines 378-380). The y-axis labeling has been corrected in all figures showing surfactant secretion (Main Figures 1c, 1e, 2a, 3f, 4c, 5e, and 7c, and Extended Data Figures 1g and h).

16.2. The pre-stretch surfactant secretion profile is given in Extended Data Figure 1g.

#17. The abstract could be rewritten to highlight more of the mechanistic details in the paper. We revised the abstract.

#18. All the statements causally link loss of barrier function to impaired surfactant secretion should be reworded or explained since the direct causal link between surfactant and barrier permeability is not immediately clear.

The statements are revised in light of new findings as we discuss in Response #2.1.

#19. LPS doses should be included on the figures. Otherwise, it's unclear why LPS sometimes causes 100% mortality in WT mice (Fig. 7F) and sometimes does not (Fig. 2H).

The difference in mortality is because we used LPS at lethal and sublethal doses, as we now indicate in the figure legends. Also, LPS doses differed with strain. In Supplementary Table 3, we have listed the LPS doses we used for different strains. Further, we revised the text throughout to clarify LPS dose and strain issues.

References

- 1 Sisson, T. H. *et al.* Expression of the reverse tetracycline-transactivator gene causes emphysema-like changes in mice. *Am J Respir Cell Mol Biol* **34**, 552-560, doi:10.1165/rcmb.2005-0378OC (2006).
- 2 Perez-Gil, J. Structure of pulmonary surfactant membranes and films: the role of proteins and lipid-protein interactions. *Biochim Biophys Acta* **1778**, 1676-1695, doi:10.1016/j.bbamem.2008.05.003 (2008).
- 3 Hough, R. F. *et al.* Endothelial mitochondria determine rapid barrier failure in chemical lung injury. *JCI Insight* **4(3)**, e124329, doi:10.1172/jci.insight.124329 (2019).

REVIEWERS' COMMENTS

Reviewer #1 (Remarks to the Author):

The authors have partially addressed my concerns. I don't feel they have provided a rigorous set of experiments delineating how LPS induces a decline in MCU expression. With that said, I take their point that his angle is perhaps outside the scope of this more translational study. There is already quite a lot of interesting data here and while I would have preferred to learn more about what was happening to MCU, I understand this might be best achieved in subsequent studies.

Reviewer #2 (Remarks to the Author):

Excellent revisions with valuable additional data, and clear additions to the discussion.

Point-by-point response to Reviewers' comments (August 8, 2022)

We thank the reviewers for their comments, which have enhanced the manuscript in revision. Our responses are indicated in bold below.

Reviewer 1:

The authors have partially addressed my concerns. I don't feel they have provided a rigorous set of experiments delineating how LPS induces a decline in MCU expression. With that said, I take their point that his angle is perhaps outside the scope of this more translational study. There is already quite a lot of interesting data here and while I would have preferred to learn more about what was happening to MCU, I understand this might be best achieved in subsequent studies.

We thank the Reviewer for the kind comments and for raising the issue of the MCU loss. We expect to pursue this question.

Reviewer 2:

Excellent revisions with valuable additional data, and clear additions to the discussion.

We thank the Reviewer for the kind comments.

References

- 1 Sisson, T. H. *et al.* Expression of the reverse tetracycline-transactivator gene causes emphysema-like changes in mice. *Am J Respir Cell Mol Biol* **34**, 552-560, doi:10.1165/rcmb.2005-0378OC (2006).
- 2 Perez-Gil, J. Structure of pulmonary surfactant membranes and films: the role of proteins and lipid-protein interactions. *Biochim Biophys Acta* **1778**, 1676-1695, doi:10.1016/j.bbamem.2008.05.003 (2008).
- 3 Hough, R. F. *et al.* Endothelial mitochondria determine rapid barrier failure in chemical lung injury. *JCI Insight* **4(3)**, e124329, doi:10.1172/jci.insight.124329 (2019).